# Dishevelled has a YAP nuclear export function in a tumor suppressor context-dependent manner

Yoonmi Lee[1,2], Nam Hee Kim[2], Eunae Sandra Cho[1], Ji Hye Yang[1], Yong Hoon Cha[3], Hee Eun Kang[1], Jun Seop Yun[1], Sue Bean Cho[2], Seon-Hyeong Lee[4], Petra Paclikova[5], Tomasz W. Radaszkiewicz[5], Vitezslav Bryja [5], Chi Gu Kang[1], Young Soo Yuk[1], So Young Cha[1], Soo-Youl Kim[4], Hyun Sil Kim[2] & Jong In Yook [1]

Phosphorylation-dependent YAP translocation is a well-known intracellular mechanism of the Hippo pathway; however, the molecular effectors governing YAP cytoplasmic translocation remains undefined. Recent findings indicate that oncogenic YAP paradoxically suppresses Wnt activity. Here, we show that Wnt scaffolding protein Dishevelled (DVL) is responsible for cytosolic translocation of phosphorylated YAP. Mutational inactivation of the nuclear export signal embedded in DVL leads to nuclear YAP retention, with an increase in TEAD transcriptional activity. DVL is also required for YAP subcellular localization induced by E-cadherin, α-catenin, or AMPK activation. Importantly, the nuclear-cytoplasmic trafficking is dependent on the p53-Lats2 or LKB1-AMPK tumor suppressor axes, which determine YAP phosphorylation status. In vivo and clinical data support that the loss of p53 or LKB1 relieves DVL-linked reciprocal inhibition between the Wnt and nuclear YAP activity. Our observations provide mechanistic insights into controlled proliferation coupled with epithelial polarity during development and human cancer.

[1] Department of Oral Pathology, Yonsei University College of Dentistry, Seoul 03722, Korea. [2] Oral Cancer Research Institute, Yonsei University College of Dentistry, Seoul 03722, Korea. [3] Department of Oral and Maxillofacial Surgery, Yonsei University College of Dentistry, Seoul 03722, Korea. [4] Cancer Cell and Molecular Biology Branch, National Cancer Center, Ilsan 10408, Korea. [5] Institute of Experimental Biology, Faculty of Science, Masaryk University, Brno 62500, Czech Republic. These authors contributed equally: Yoonmi Lee, Nam Hee Kim. Correspondence and requests for materials should be addressed to H.S.K. (email: khs@yuhs.ac) or to J.I.Y. (email: jiyook@yuhs.ac)

The Hippo signaling is an evolutionary conserved pathway that inhibits cell proliferation by contact inhibition, its loss leading to both organ growth and cancer development. The Yes-associated protein (YAP) transcription co-activator is a key regulator of the Hippo pathway[1,2]. Inhibition of the Hippo pathway leads to increased nuclear YAP abundance and TEAD transcriptional activity, resulting in increased organ size as well as overgrowth of cancer[3,4]. Conversely, activation of the Hippo pathway induced by cell-to-cell contact leads to phosphorylation and inhibition of nuclear YAP. In mammals, large tumor suppressor (Lats)1/2 serine/threonine kinase phosphorylates at multiple sites on YAP, including Ser127, resulting in cytoplasmic translocation from the nucleus[1,5,6]. Recently, AMP-activated protein kinase (AMPK) has been shown to directly phosphorylate YAP, resulting in cytoplasmic retention and suppression of nuclear YAP activity[7,8]. While the phosphorylation-dependent YAP shuttling is critically important in the Hippo pathway and/or in metabolic regulation, the molecular effector of the dynamic intracellular shuttling is not known.

The canonical Wnt pathway comprises fundamental extracellular signaling involving diverse developmental process, and deregulation of components involved in the Wnt/β-catenin pathway has been implicated in a wide spectrum of diseases, particularly human cancers[9]. Highly conserved in metazoan, the Wnt signaling is critically important for coordinative regulation of cell-to-cell adhesion from cell membrane to transcriptional activity in the nucleus. The β-catenin, a key mediator of Wnt signaling, functions both as intercellular adhesion complex through binding to cytoplasmic domain of E-cadherin and as transcriptional co-activator in the nucleus with T-cell factor/lymphoid enhancer factor (TCF/LEF)[9,10]. Because the Hippo and Wnt pathways similarly regulate intercellular adhesion and nuclear transcriptional activity[11], elucidating a reciprocal link between the two pathways may reveal an important molecular mechanism in human cancer and other diseases. Although the co-activation of Wnt signaling and YAP activity are commonly observed in human cancer, recent findings point to a dilemma in that YAP suppresses canonical Wnt via binding to Disheveled (DVL) and/or β-catenin[2,12–15]. Although a large body of studies have focused on YAP regulation of canonical Wnt activity in development and cancer[12–14], the upstream function and molecular mechanisms enabling reciprocal regulations between YAP and Wnt signaling are largely unknown[16].

In this study, we found that DVL, a scaffolding protein of the Wnt pathway as well as a key regulator of Wnt-independent epithelial polarity, is a molecular effector for nuclear-cytoplasmic shuttling of YAP in a YAP phosphorylation-dependent manner. Furthermore, oncogenic inactivation of p53/Lats2 and the liver kinase B1 (LKB1)/AMPK tumor suppressor axes, two most commonly observed genetic alterations in human cancer, abolish DVL's function on YAP nuclear export. The loss of tumor suppressor function allows co-activation of the canonical Wnt pathway and nuclear YAP activity by DVL. Our observations demonstrate molecular mechanisms for the dynamic regulation of YAP activity via subcellular trafficking by DVL as well as the importance of p53 and LKB1 tumor suppressor contexts in the reciprocal control between the canonical Wnt and Hippo pathways.

## Results

### DVL interacts with YAP in a phosphorylation-dependent manner.
Because the YAP antagonizes Wnt activity via binding to DVL in development and human cancer[2,13], we focused on roles of enigmatic DVL on YAP activity in this study. As a key scaffolding protein of the Wnt pathway, DVL in mammal consists of three highly similar homolog genes, *DVL1*, *DVL2*, and *DVL3*. Although DVL2 has received close attention in recent developmental studies due to its ubiquitous abundance in various tissues[17–19], differential expression patterns of DVL homologs in human cancer have not been clearly determined[20]. We first examined the transcript abundance of DVL homologs from the 1093 breast cancer patients and found that DVL3 transcripts were most abundant in clinical samples (Supplementary Fig. 1a). Because the stability and protein level of DVL are controlled by post-translational modification with many DVL-interacting proteins[21], we next examined relative abundance of transcripts and protein in human cancer cells. Consistently, DVL3 is most abundant in the human cancer cell lines panel (Supplementary Fig. 1b, c), indicating that DVL3 is mainly expressed in human cancer. To examine crosstalk between Wnt and YAP, we next explored interactions between DVL3 and YAP. Consistent with previous observations[12,13], immunofluorescence assay with endogenous DVL3 and YAP revealed that those proteins are co-localized mainly in cytoplasm under confluent condition in MCF-10A and MCF-7 epithelial cells (Fig. 1a and Supplementary Fig. 1d). Because the phosphorylation on multiple sites of YAP plays key roles in its cytosolic localization[5,6], we examined potential interactions of DVL based on YAP phosphorylation status. Co-immunoprecipitation assay with epitope-tagged proteins revealed that wild-type (wt) YAP interacted with DVL3, whereas phosphorylation-resistant mutants of YAP largely abolished the interaction (Fig. 1b). To simply examine YAP phosphorylation-dependent interaction, we treated lambda protein phosphatase (λ PPase) in vitro to immunoprecipitated YAP and subjected it to DVL binding. The λ PPase treatment reduced YAP phosphorylation status and ablated its DVL binding (Fig. 1c). The Lats1/2 kinases and AMPK are well-known kinases regulating YAP phosphorylation[5–8]. To further determine the roles of these kinases in the DVL-YAP interaction, we introduced the dominant negative mutants of Lats2 (Lats2-KR), kinase-dead AMPK (AMPK-KD) or LKB1 mutant (LKB1-KD) with DVL3. Indeed, the kinase-dead dominant negative mutant of Lats2 or AMPK or LKB1 attenuated DVL-YAP interaction with decreasing YAP phosphorylation as determined by mobility shift on a phos-tag gel and pSer127-specific YAP antibody (Fig. 1d). In physiological condition of epithelial cells, it is well-known that contact inhibition of epithelial cells increases YAP phosphorylation. To examine interaction of endogenous YAP phosphorylation on DVL, we next compared the YAP phosphorylation status under sparse and confluent state of MCF-10A cells, and subjected it for immunoprecipitation assay. Indeed, contact inhibition under confluent state increased endogenous YAP phosphorylation and its subsequent binding to DVL3 (Fig. 1e). TEAD (TEA domain family member) transcription factor functions to retain YAP in nucleus, while 14-3-3 inhibits nuclear YAP activity via cytoplasmic retention of phosphorylated YAP[2]. We, therefore, compared the ability of DVL to bind TEAD or 14-3-3, and found that DVL interacted with 14-3-3 (Supplementary Fig. 2a). However, DVL did not bind to TEAD while the TEAD interacted with YAP, which served as a positive control for TEAD binding (Supplementary Fig. 2b). In an immunofluorescence study, TEAD and DVL3 were differentially localized in nucleus and cytoplasm, respectively. These results indicate that DVL mainly interacts with cytoplasmic YAP in a phosphorylation-dependent manner.

### DVL suppresses YAP nuclear abundance and TEAD activity.
Although recent studies have focused on the role of YAP in antagonizing Wnt activity, the upstream functions by which the Wnt scaffolding protein affect nuclear YAP and TEAD transcriptional activity have not been widely studied. To address the

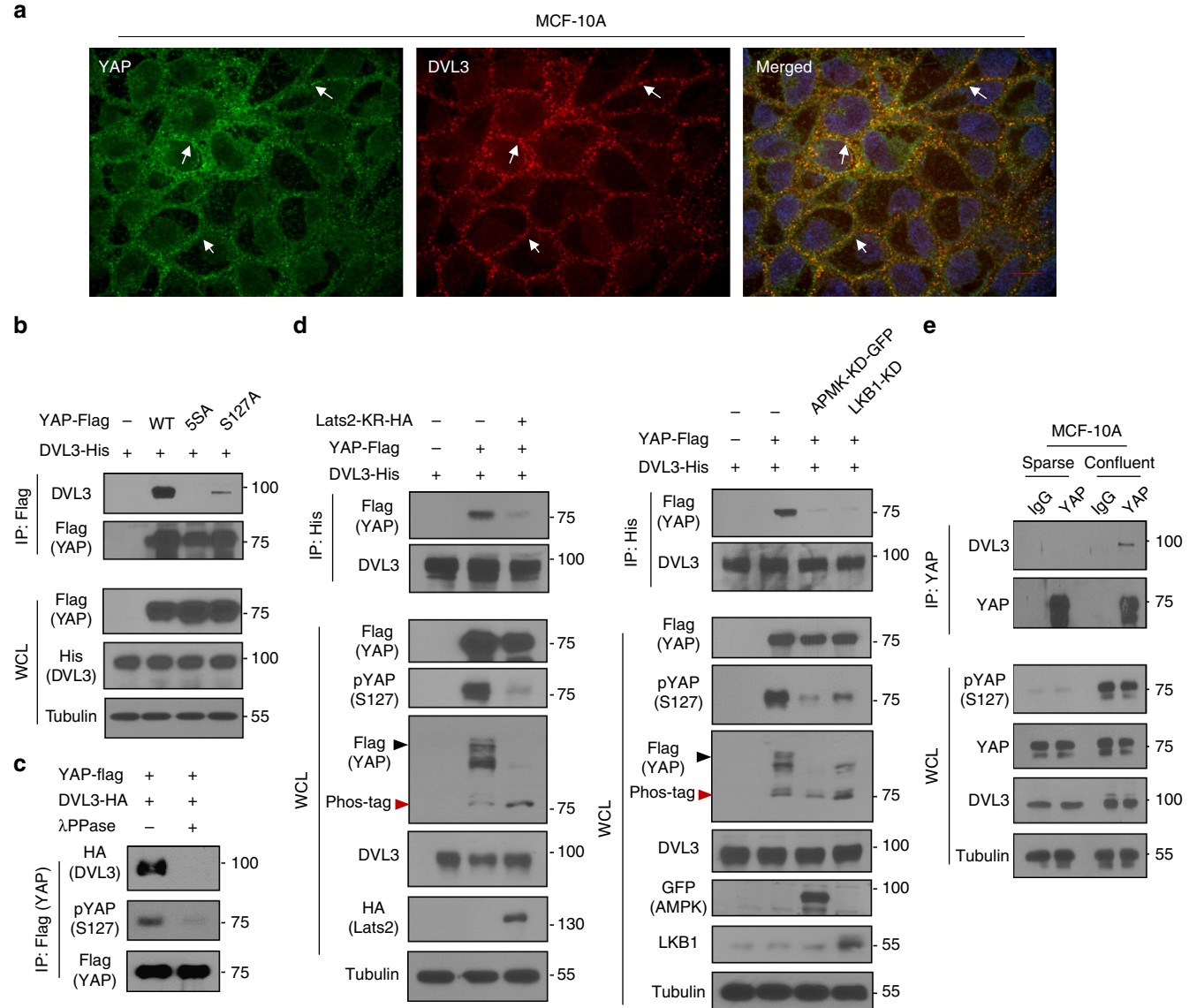

**Fig. 1** DVL interacts with phosphorylated YAP. **a** Confocal images of endogenous YAP (green) and DVL3 (red) in MCF-10A cells. Arrows indicate co-localized foci. Nuclear staining with TOPRO3 (blue) is shown in merged image. Scale bar, 10 μm. **b** DVL interacts with YAP in a phosphorylation-dependent manner. In all, 293 cells were transfected with His-tagged DVL3 and vector control (−) or flag-tagged YAP or mutants (5SA, S127A). Interactions between DVL and YAP were determined following immunoprecipitation (IP) with anti-flag antibody and immunoblotting with anti-HA. Whole-cell lysate (WCL) serves as input abundance for IP. **c** Lambda protein phosphatase (λ PPase) treatment to immunoprecipitated YAP abolishes DVL binding. The 293 cells were transfected with flag-tagged YAP and immunoprecipitated anti-flag beads were treated with λ PPase (+). The agarose beads were then subjected to binding to HA-tagged DVL. **d** Kinase-dead dominant negative Lats2 (Lats2-KR) or AMPK (AMPK-KD) or LKB1 (LKB1-KD) abolishes YAP and DVL interaction. Flag-tagged YAP and His-tagged DVL3 expression vectors were co-transfected with the dominant negative expression vectors as indicated in 293 cells. Interactions between DVL3 and YAP were determined as described above and phosphorylation status of YAP was determined by pS127-YAP antibody and mobility shift on a phos-tag gel. Black and red arrowheads correspond to the fully phosphorylated and active YAP on a phos-tag gel, respectively. **e** The MCF-10A cells were cultured under sparse and confluent states, and the whole-cell lysates (WCL) were subjected for immunoblot analysis and immunoprecipitation (IP) assay with anti-YAP antibody. Mouse IgG served as negative control. Unprocessed original scans of blots are shown in Supplementary Fig. 10

upstream role of DVL, we overexpressed DVL3 and measured TEAD transcriptional activity with a synthetic reporter construct containing multimerized responsive elements of TEAD. Interestingly, DVL3, as well as DVL1 or DVL2, significantly suppressed TEAD transcriptional activity although the total protein abundance of YAP increased slightly in MCF-7 and 293 cells (Fig. 2a and Supplementary Fig. 3a, b). Similarly, DVL3 suppressed TEAD transcriptional activity and transcript abundance of CTGF (Fig. 2a), a representative transcription target of the YAP/TEAD complex, suggesting that DVL inhibits nuclear YAP

transcriptional activity in cells. Given observations that DVL interacts with phosphorylated YAP, we hypothesized that DVL controls nuclear YAP activity via intracellular dynamics rather than by regulating YAP abundance. To prove this notion directly, cells were transfected with DVL3 expression vector and the nuclear YAP compartmentalization was assessed. Indeed, DVL3 significantly depleted nuclear YAP abundance in 293 and MCF-7 cells (Fig. 2b). The YAP has multiple phosphorylation sites involving cytoplasmic retention and protein stability of YAP[5,6], and Ser127 phosphorylation is required for cytoplasmic

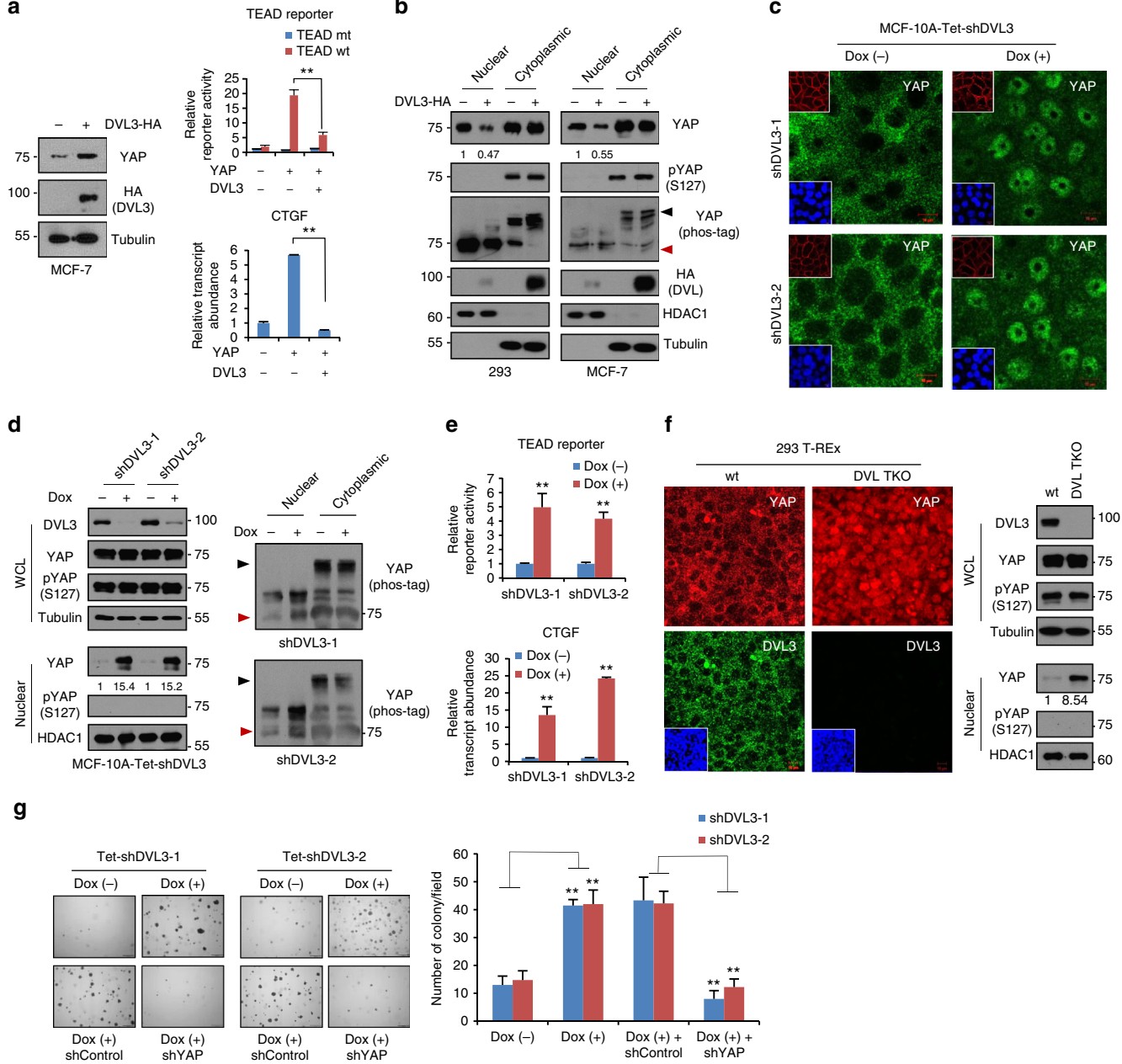

**Fig. 2** DVL suppresses YAP nuclear abundance and TEAD transcriptional activity. **a** YAP was co-transfected with HA-DVL3 into MCF-7 cells, and protein abundance (left), TEAD reporter activity (right upper), and CTGF transcript abundance (right lower) were determined. Data of reporter assay and RT-PCR are normalized to negative control empty vector (−) and presented as mean ± SD. **b** The 293 and MCF-7 cells were transfected with DVL3, and protein abundance of YAP in nuclear and cytoplasmic fraction was determined by immunoblot analysis. YAP phosphorylation status in cytoplasmic and nuclear fraction was determined by pS127-YAP antibody and mobility shift on a phos-tag gel. Red arrowhead indicates active YAP on a phos-tag gel. Tubulin and HDAC1 served as loading controls of cytosolic fraction and nuclear lysates, respectively. Relative nuclear YAP abundance compared to control was measured by ImageJ. **c**, **d** Inducible knockdown of DVL3 increases nuclear YAP abundance. The MCF-10A cells expressing tetracycline-inducible shRNA against DVL3 were generated with lentiviral system, and endogenous YAP localization and abundance without or with doxycycline (Dox) were determined by immunofluorescence (**c**) and immunoblot analysis from whole-cell lysate (WCL) and nuclear fraction (**d**). The cells were serum-starved for 16 h before harvest. Tubulin and HDAC1 served as loading controls of cytosolic fraction and nuclear lysates, respectively. **e** DVL3 was knockdowned with doxycycline (Dox), and the TEAD reporter activity (upper panel) and CTGF transcript abundance (lower panel) were determined by reporter assay and qRT-PCR, respectively. **f** The wt and DVL-TKO cells were cultured in confluent condition and subcellular localization of endogenous YAP and DVL3 were determined by confocal microscopy (left panels) and immunoblot analysis (right panels). The cells were serum-starved for 16 h before examination. Inset, DAPI nuclear stain; Scale bar, 10 μm. **g** Knockdown of DVL3 increases anchorage-independent growth of MCF-7 cells. The MCF-7 cells expressing inducible shRNA for DVL3 were seeded onto a soft agar without (Dox-) or with (Dox+) doxycycline in combination with shControl or shYAP for 3 weeks. The colonies were stained with crystal violet and quantified. Data presented as mean ± SD, $n = 5$

translocation of YAP[5]. Consistently, nuclear YAP was unphosphorylated regardless of DVL3 abundance as determined by pSer127-YAP antibody and mobility shift on a phos-tag gel. TAZ (transcriptional co-activator with a PDZ-binding motif), YAP homolog and an integral member of the Hippo pathway, binds DVL[12]. When we examined the effect of DVL on TAZ, the DVL also interacted with TAZ and suppressed nuclear TAZ abundance, TEAD transcriptional activity indicating the critical function of DVL on the Hippo pathway (Supplementary Fig. 3c). To further determine the role of endogenous DVL, we made inducible shRNA constructs against DVL3 and examined nuclear YAP abundance. The cytoplasmic translocation of YAP by contact inhibition is a well-known intracellular mechanism of the Hippo pathway[5,6]. When we cultured confluent epithelial MCF-10A or MCF-7 cells, contact inhibition led to cytoplasmic translocation of YAP while inducible knockdown of DVL3 largely increased its nuclear retention (Fig. 2c and Supplementary Fig. 4a). Interestingly, the nuclear YAP remained unphosphorylated although nuclear YAP abundance was increased by the loss of endogenous DVL3 (Fig. 2d). These results indicate that endogenous DVL is required for cytoplasmic translocation of Ser127 phosphorylated YAP and nuclear YAP remains active regardless of DVL abundance. DVL's role in YAP nuclear retention was functional in terms of TEAD reporter activity and CTGF transcript abundance (Fig. 2e). To unambiguously establish DVL as a critical effector of YAP trafficking, we next used wt and DVL1/2/3-triple knockout (DVL-TKO) 293 T-REx cells to examine endogenous YAP localization[22]. In every case, YAP largely remained in the nuclear space in DVL-TKO cells under confluent contact inhibition while the YAP phosphorylation status was unchanged by knockout of DVLs (Fig. 2f). Reintroduction of wt DVL3 into DVL-TKO cells successfully rescued YAP translocation into cytoplasmic space (Supplementary Fig. 4b). The effect of nuclear YAP on oncogenic transformation having been clearly demonstrated by anchorage-independent growth assay[6,8], we thus next examined the effect of such DVL-mediated nuclear YAP on oncogenic transformation. Knockdown of DVL3 was sufficient to increase the transforming potential of MCF-7 cells in a YAP-dependent manner (Fig. 2g). These observations indicate that DVL regulates nuclear abundance and transcriptional activity of YAP.

**DVL enables YAP nuclear export.** To elucidate the mechanistic link between DVL and nuclear YAP activity, we next examined the influence of DVL's conserved domains on YAP interaction. The DVL has several conserved domains, an N-terminal DIX, a central PDZ, and a C-terminal DEP, all implicated in mediating many cellular functions with variable interacting partners[23]. To test whether these domains are responsible for nuclear YAP activity, we made deletion constructs and tested the ability of DVL mutants to redirect YAP localization. However, deletion mutants of those conserved domains suppressed nuclear YAP abundance and TEAD transcriptional activity, the deletion mutants of DVL3 retaining binding ability to YAP (Supplementary Fig. 5a). In the C-terminus of YAP, there is a PDZ-binding motif required for interaction with YAP binding proteins having the PDZ domain[24]. Indeed, deletion of the PDZ-binding motif ablated YAP binding to DVL (Supplementary Fig. 5b). SET7 (SETD7)-mediated monomethylation of lysine 494, located close to the PDZ-binding domain, has been identified as critical for cytoplasmic localization of YAP despite S127 phosphorylation[25]. To determine the role of lysine monomethylation, we made a K494R mutant of YAP and subjected it to DVL interaction. Interestingly, a point mutant of the lysine was sufficient to abolish DVL interaction (Supplementary Fig. 5b), indicating that the

PDZ-binding domain and monomethylation may cooperate for DVL interaction. The YAP WW domains interact with multiple proteins, such as AMOT and ERBB4[26,27]. To examine whether the WW domains play a role in DVL interaction, we made a deletion mutant of WW domains and subjected it to immuno-precipitation assay. Interestingly, deletion of the WW domains largely abolished DVL interaction with YAP (Supplementary Fig. 5b). These results indicate that both WW domain and the PDZ-binding domain are involved in DVL interaction. Given that the PDZ domain of DVL is necessary, but not sufficient, for YAP interaction, we next examined DVL domains for YAP interaction. Because TAZ interacts with the PY motif (PPxY) as a WW domain-binding ligand and PDZ domain of DVL[12,28], we next made deletion mutants of the PY motif and PDZ domain of DVL to examine the interaction of the mutants with YAP. Like TAZ, the PDZ domain and a PY motif of DVL both contribute to YAP interaction (Supplementary Fig. 5b). Note that deletion of the PY motif and the PDZ domain of DVL unable to bind to YAP did not translocate YAP into cytoplasm (Supplementary Fig. 5c), indicating that interaction with YAP is required for DVL's role in YAP subcellular localization.

Scaffolding proteins in the Wnt pathway such as APC and Axin have a nuclear export function[29–31]. Although DVL has been reported to undergo nuclear-cytoplasmic shuttling of β-catenin[32], its role in YAP intracellular trafficking has not yet been determined. To test whether DVL regulates nuclear YAP dynamics, we treated Leptomycin B (LMB), a specific inhibitor of CRM1 (chromosomal region maintenance)/exportin 1 required for nuclear export of proteins containing a leucine-rich nuclear export sequence (NES). Under confluent contact inhibition state, endogenous YAP mainly localized in the cytoplasm together with DVL (Fig. 3a). Interestingly, LMB treatment in confluent MCF-7 cells led to nuclear retention of endogenous DVL and YAP together, supporting that the nuclear export function of DVL may regulate the nuclear-cytosolic dynamics of YAP. Interestingly, the DVL family amino-acid sequence contains highly conserved typical NES consisting of M/LxxLxL (capital letter Met or Leu amino acids, where x is any amino acid) next to the DEP domain (Fig. 3b)[32]. To test whether the NES of DVL is responsible for intracellular YAP shuttling, we made a point mutant of DVL3 whose Leu in this NES was substituted with Ala (ASA mutant). When we examined the subcellular localization of wt or ASA point mutant DVL3, mutational inactivation of NES was sufficient to restrict DVL in nucleus (Supplementary Fig. 5d), indicating that NES is critically required for cytoplasmic translocation of DVL. We then made inducible wt and ASA mutant of DVL in MCF-10A and MCF-7 cells to assess for endogenous YAP compartmentalization with doxycycline treatment. Whereas DVL and YAP were mainly co-localized in the cytoplasm in inducible wt DVL3 cells, the induction of ASA mutant led to distinctive nuclear retention of DVL3 and YAP together (Fig. 3c and Supplementary Fig. 5e). When we examined nuclear YAP level with an inducible DVL3 system, nuclear abundance of endogenous YAP was not suppressed by the ASA mutant DVL3 in MCF-7 and 293 cells (Fig. 3d). As with knockdown of DVL3, induction of ASA mutant did not affect nuclear YAP phosphorylation status. Given DVL's interaction with phosphorylated YAP, we next examined whether ASA mutant of DVL interacts with unphosphorylated YAP in nuclear space. Interestingly, NES-mutant of DVL did not interact with nuclear YAP (Fig. 3e). To examine whether the disruption of DVL's nuclear export is functional on YAP transcriptional activity, we determined TEAD reporter activity and CTGF transcript abundance. Consistent with increased active YAP in nuclear space, induced expression of ASA mutant increased TEAD reporter activity and CTGF transcript level in MCF-7 cells

(Fig. 3f). To assess the functional importance of activation of nuclear YAP by ASA mutant, we next performed anchorage-independent growth assay. Consistently, YAP increased anchorage-independent growth of MCF-7 cells and DVL suppressed YAP-mediated soft agar growth (Fig. 3g). The ASA mutant of DVL3 rescued the wt DVL function on YAP-mediated

anchorage-independent growth. To validate nuclear export of YAP by DVL in vivo, we next made tumors of 293 cells expressing inducible wt or ASA mutant DVL3 and examined YAP localization from xenografted tissue. Indeed, YAP was mainly localized in cytoplasmic and nuclear space by induction of wt or ASA mutant DVL3, respectively, (Fig. 3h). Thus, the

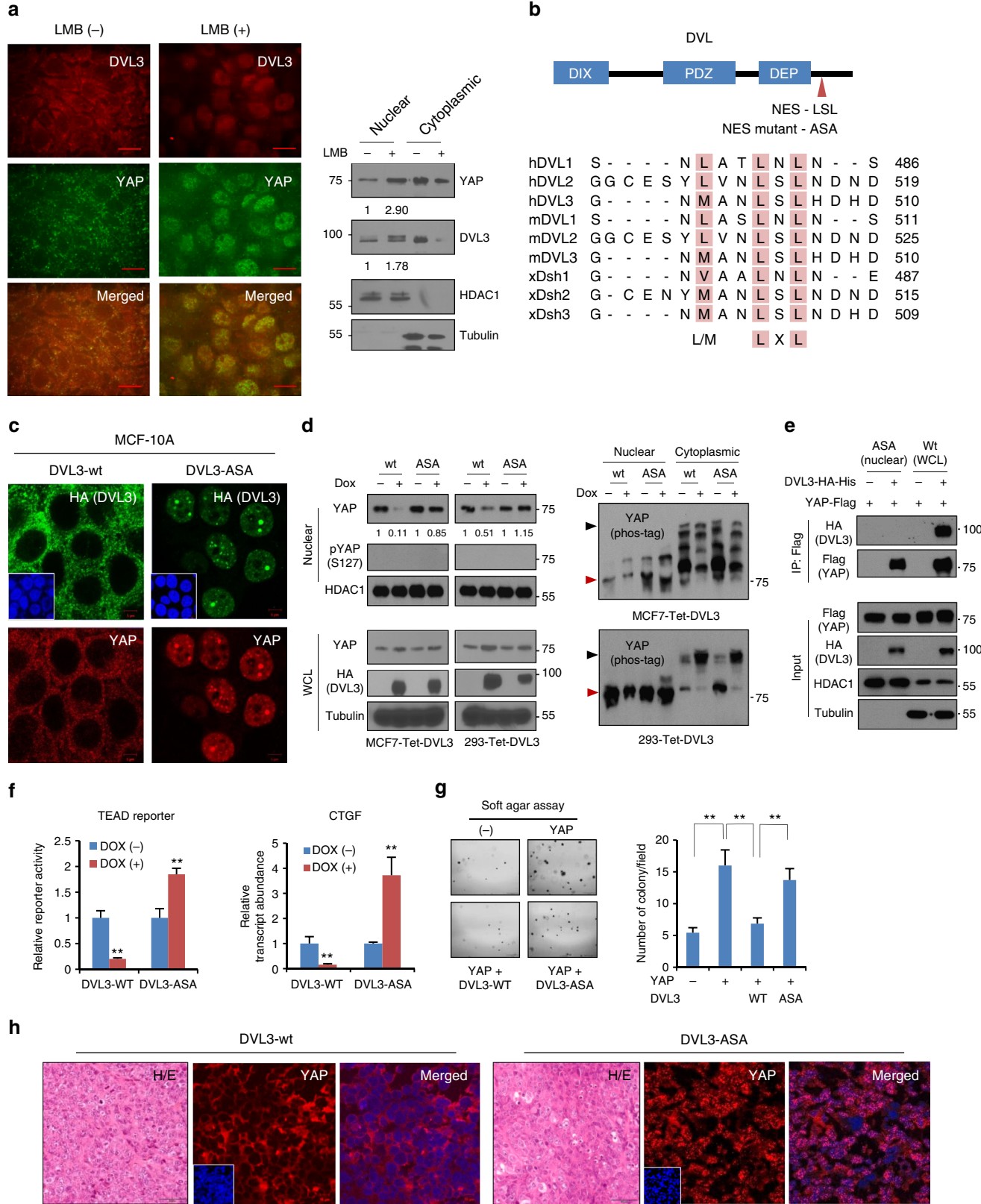

nuclear export function of DVL is responsible for intracellular trafficking and YAP transcriptional activity.

**Phosphorylation-dependent YAP nuclear export by DVL**. Given the YAP phosphorylation-dependent binding of DVL, we next examined how YAP phosphorylation status affected DVL's nuclear export function. To examine the effect of YAP phosphorylation on DVL's function, we next assessed DVL suppression of YAP phosphorylation-resistant mutants. Indeed, the suppressor role of DVL3 by means of TEAD reporter activity and nuclear YAP abundance was largely abolished in the YAP phosphorylation-resistant mutants (Fig. 4a). In an immuno-fluorescence study, wt YAP consistently co-localized with DVL3 mainly in cytoplasm (Fig. 4b). However, the phosphorylation-resistant mutants of YAP were localized in nuclear space independent of DVL3. Because the Lats1/2 are well-known kinases of YAP phosphorylation at multiple sites, we transfected the kinase-dead dominant negative Lats2 and examined the extent to which DVL suppressed TEAD transcriptional activity. Interestingly, dominant negative Lats2 largely abolished DVL suppression of YAP transcriptional activity (Fig. 4c), suggesting that the nuclear export function of DVL depends on YAP phosphorylation. To unambiguously establish Lats kinases as critical regulators of DVL-mediated YAP translocation, we used Lats1/2 double-knockout (Lats1/2$^{-/-}$) mouse embryonic fibroblast (MEF) and 293A cells. When we induced DVL3 in these cells, the nuclear YAP abundance in the Lats1/2$^{-/-}$ cells were not decreased by DVL3 (Fig. 4d and Supplementary Fig. 6). Therefore, DVL, a well-known scaffolding protein of the Wnt pathway and a regulator of epithelial cell polarity, is a molecular effector of the Hippo pathway regulating nuclear export of phosphorylated YAP (Fig. 4e).

Wnt ligands, essential morphogens in receptor-mediated signaling pathways, control development and tissue homeostasis. Hyperactivation of the Wnt pathway is frequently observed in many types of human cancer[33,34]. Recent studies have uncovered the interaction between YAP/TAZ and Wnt signaling, with DVL emerging as the hub that integrates YAP and Wnt[12,13]. Intriguingly, Wnt ligands promote YAP activation via the alternative Wnt pathway in a Lats-dependent manner[35]. To examine the role of DVL in Wnt-mediated YAP regulation, we treated soluble Wnt ligands and examined the YAP phosphorylation status. Indeed, treatment with Wnt1 and Wnt3a ligands increased active YAP resulting from decreased YAP phosphorylation (Fig. 4f). Consistent with YAP phosphorylation-dependent binding to DVL, soluble Wnt ligands treatment decreased DVL-YAP binding. Examining endogenous YAP and DVL localization, we found that soluble Wnt1 and Wnt3a ligands

induced nuclear YAP localization while the endogenous DVL3 largely remained in cytoplasmic space (Fig. 4g). These results suggest that Wnt ligand activates YAP by inhibiting YAP phosphorylation, resulting in nuclear translocation of YAP subsequent to its release from DVL.

**DVL's role for YAP localization by adherens junction**. During the contact inhibition of epithelial cells, the E-cadherin/α-catenin complex, well-known members of the adherens junction, plays a critical role in the phosphorylation and cytosolic translocation of YAP via sensing cell contact[36–38]. To test whether DVL is required for E-cadherin or α-catenin-induced YAP translocation, we next assessed the role of DVL in YAP translocation induced by E-cadherin or α-catenin. We depleted endogenous DVL using DVL-TKO 293 T-REx cells or shRNA-mediated knockdown of DVL3. In an immunofluorescence study, E-cadherin or α-catenin translocated YAP into cytoplasm in wt cells while loss of endogenous DVL in DVL-TKO cells or by shRNA-mediated knockdown largely retained YAP in nuclear space (Fig. 5a and Supplementary Fig. 7a). Consistently, overexpression of E-cadherin or α-catenin decreased TEAD reporter activity and nuclear YAP abundance while loss of DVL largely abolished those effects (Fig. 5b and Supplementary Fig. 7b). To further examine the contribution of DVL induced by the endogenous adherens junction complex, we next used a function-blocking antibody (HECD) to disrupt E-cadherin homophilic binding in DVL-inducible MCF-10A cells. As shown earlier, contact inhibition under confluent culture condition led to YAP cytoplasmic retention and functional blocking of E-cadherin with HECD-induced YAP translocation into nuclear space, while inducible DVL3 largely abolished the HECD effect on YAP (Fig. 5c). These results support the critical role of DVL in cytoplasmic translocation of YAP regulated by E-cadherin/α-catenin complex.

**Loss of p53/Lats axis relieves YAP restriction by DVL**. Genetic analysis of DVL in a developmental system has shown it to be a potent activator of canonical Wnt signaling[39]. While YAP is regarded as an oncogene in many cancers, recent observations have indicated a tumor suppressor role of cytosolic YAP by restriction of canonical Wnt[12–14]. To resolve this paradox, we next examined the effects of upstream signals of YAP phosphorylation with respect to DVL's nuclear export function. Previously, Lats2 has been identified as a direct transcriptional target of p53[40], suggesting that the p53/Lats2 tumor suppressor axis provides a context for reciprocal regulation of canonical Wnt and YAP by DVL. To test this idea, we knockdowned wt p53 function and examined DVL's affect on nuclear YAP and canonical Wnt activity. Indeed, knockdown of p53 using shRNA or HPV

**Fig. 3** NES (nuclear export sequence) in DVL is responsible for YAP trafficking. **a** The confluent MCF-7 cells were treated with Leptomycin B (LMB, 5 ng ml$^{-1}$) for 4 h, and endogenous YAP and DVL localization and abundance were determined by confocal microscopy (left panels) and immunoblotting (right panels). Scale bar, 10 μm. **b** Schematic representation of the conserved NES and point mutant (NES-mutant-ASA) in DVL of human (h), mouse (m), and xenopus (x). **c** The MCF-7 cells were transfected with wt or NES-ASA mutant DVL3 and DVL-YAP localization was determined by confocal microscopy. The cells were serum-starved for 16 h before harvest. Scale bar, 5 μm. **d** MCF-7 and 293 cells expressing inducible HA-tagged wt or NES-mutant (ASA) of DVL3 were treated with doxycycline, and YAP and DVL abundance were determined (left panels). YAP phosphorylation status in cytoplasmic and nuclear fraction was determined by pS127-YAP antibody and mobility shift on a phos-tag gel (right panels). Red arrowhead indicates active YAP. **e** The 293 cells were transfected with flag-tagged YAP with NES-mutant or wt DVL3. The nuclear protein form NES-mutant transfectant was subjected for immunoprecipitation, whole-cell lysate of wt DVL serving as control. Fifty micro grams of nuclear protein and 10 μg of whole-cell lysates were used for immunoprecipitation assay to adjust YAP abundance. **f** TEAD reporter activity and CTGF transcript abundance were analyzed from MCF-7 cells expressing inducible wt or NES-mutant of DVL3 (mean ± SD, n = 3). **g** The MCF-7 cells were transfected with YAP in combination with wt or NES-ASA mutant of DVL3, then seeded onto a soft agar for 3 weeks. The colonies were stained with crystal violet and quantified (mean ± SD, n = 5). **h** The 293 cells stably expressing tet-inducible wt or NES-mutant of DVL were injected into athymic nude mice subcutaneously. When the tumor volume reached around 500 mm$^3$ (n = 1), the mice were treated with doxycycline (50 mg/kg) intraperitoneally prior sacrifice to 24 h. The tissues were examined by H/E staining (scale bar, 50 μm) and YAP localization was determined from frozen sections (scale bar, 10 μm). Unprocessed original scans of blots are shown in Supplementary Fig. 10

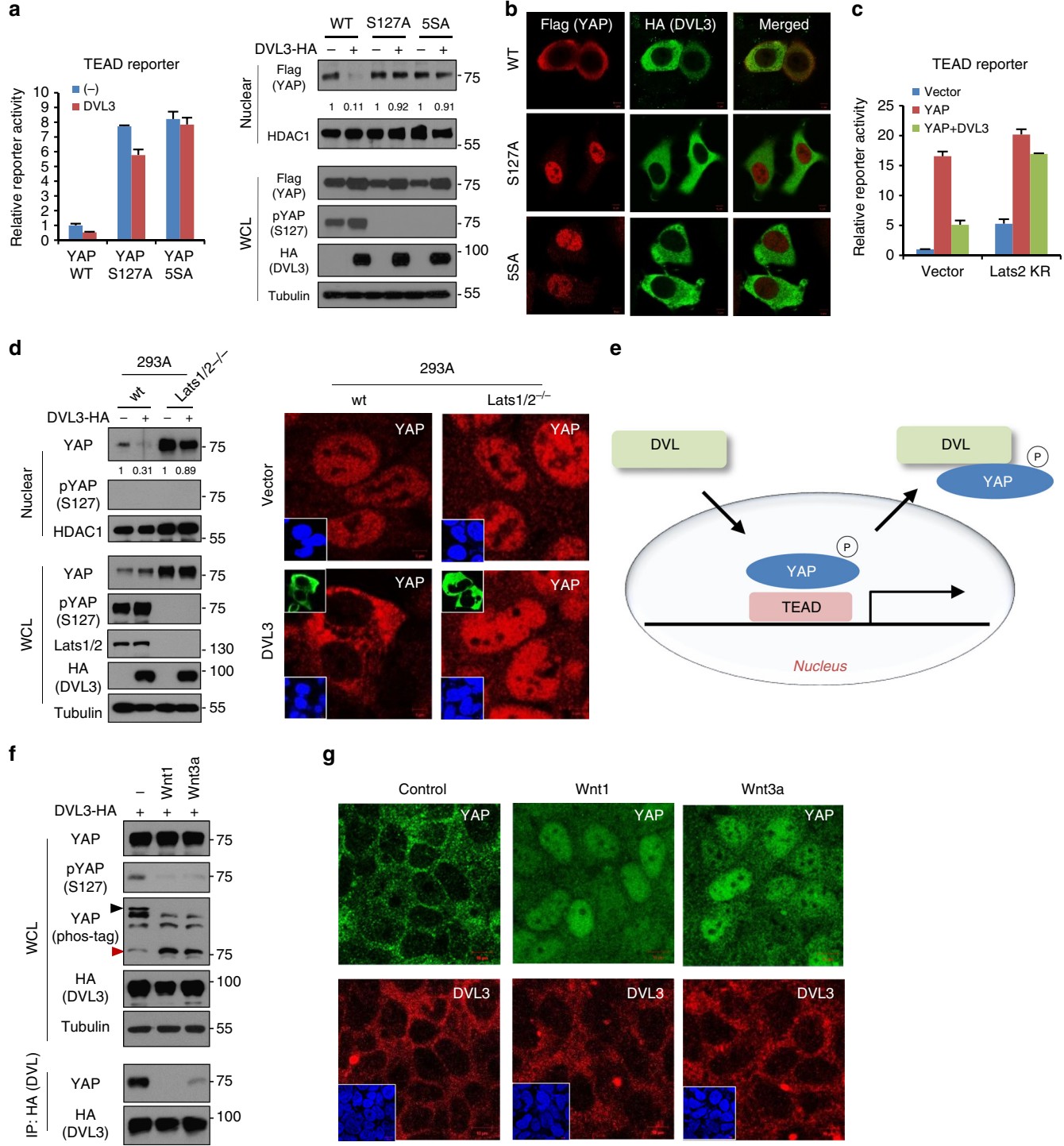

**Fig. 4** DVL has nuclear export function on phosphorylated YAP. **a** Relative fold repression of reporter activities and nuclear YAP abundance by DVL on wt or phospho-resistant mutants of YAP (S127A, 5SA) were measured with TEAD reporter assay (left panel) and immunoblot analysis (right panels), respectively. Relative nuclear YAP abundance compared to control was measured by ImageJ. **b** The MCF-7 cells were transfected with HA-tagged DVL3 in combination with flag-tagged wt or phospho-resistant mutants YAP (S127A, 5SA), and nuclear localizations of YAP and DVL were examined by confocal immunofluorescence microscopy. To minimize the overexpression issue, 10 ng of YAP expression vectors was used. Scale bar, 10 μm. **c** YAP and DVL3 were co-transfected in combination with vector control or dominant negative Lats2 (Lats2-KR), and relative TEAD reporter activity was measured. Data presented as mean ± SD, $n = 3$. **d** DVL3 was transfected in wt or Lats1/2 double-knockout (Lats1/2$^{-/-}$) 293A cells, and nuclear YAP abundance by DVL3 was determined by immunoblot analysis (left panels) and immunofluorescence study (right panels). The cells were serum-starved for 16 h before harvest. **e** Schematic diagram of nuclear export of phosphorylated YAP by DVL. **f** Wnt ligands activate YAP resulting in decreased interaction with DVL. The 293A cells transfected with HA-tagged DVL3 were serum-starved and then stimulated by Wnt1 and Wnt3a ligands for 4 h. The whole-cell lysates (WCL) were subjected for immunoblot analysis and immunoprecipitation (IP) assay with anti-HA antibody. **g** Soluble Wnt1 and Wnt3a induce YAP nuclear localization. The confluent 293A cells were serum-starved and then stimulated by Wnt ligands for 4 h. Endogenous YAP and DVL localization were determined by confocal immunofluorescence microscopy. Scale bar, 10 μm. Unprocessed original scans of blots are shown in Supplementary Fig. 10

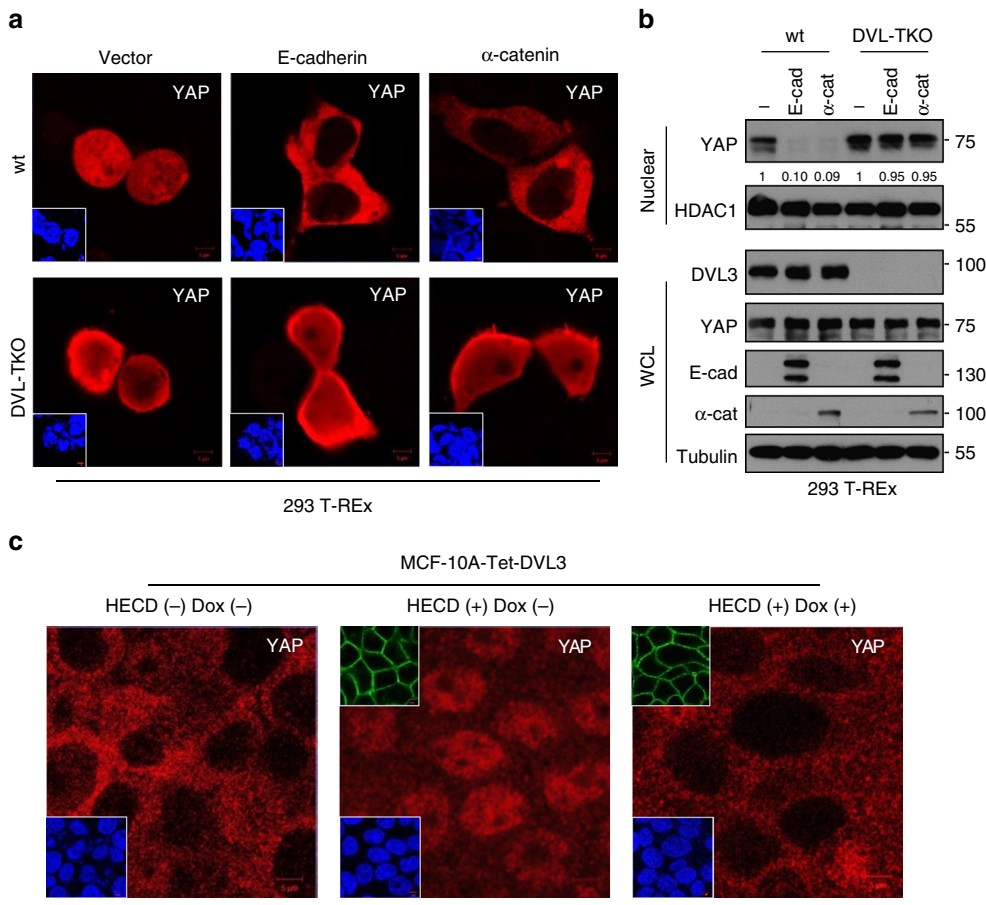

**Fig. 5** DVL is required for cytoplasmic trafficking of YAP induced by E-cadherin or α-catenin. **a** The wt and DVL-TKO 293 T-REx cells were transfected with flag-tagged YAP (50 ng) in combination with vector control (1 μg) or E-cadherin (1 μg) or α-catenin (1 μg), and YAP localization was determined by confocal microscopy. Inset, DAPI nuclear stain; Scale bar, 5 μm. **b** The wt and DVL-TKO 293 cells were transfected with vector control (−) or E-cadherin (E-cad) or α-catenin (α-cat), and endogenous YAP abundance in nuclear fraction and whole-cell lysates (WCL) was determined by immunoblot analysis. Relative nuclear YAP abundance compared to control was measured by ImageJ. **c** The MCF-10A cells were cultured under confluent contact inhibition and intracellular YAP localization was monitored by confocal microscopy. The cells were incubated with a mouse IgG (HECD-) or a neutralizing monoclonal antibody (HECD, 10 μg/ml) that disrupts homophilic binding of E-cadherin for 16 h. Immunofluorescence images showing YAP staining of MCF-10A cells having tetracycline-inducible DVL3 in the absence (Dox-) or presence (Dox+) of doxycycline. Upper inset, HECD antibody detected by fluorescent-conjugated secondary antibody; lower Inset, DAPI nuclear stain; Scale bar, 5 μm. Unprocessed original scans of blots are shown in Supplementary Fig. 10

(human papilloma virus)-E6 resulted in suppression of Lats2 protein abundance and a subsequent decrease in YAP phosphorylation (Fig. 6a and Supplementary Fig. 8a). Intriguingly, the YAP nuclear export function of DVL3 was largely abolished in those p53-loss contexts. When we examined TEAD transcriptional activity, p53 tumor suppressor context was also required for DVL to regulate nuclear YAP activity (Fig. 6b). Importantly, loss-of-p53 context still allowed DVL to potentially affect TCF/LEF transcriptional activity[41] (Fig. 6c).

We next examined the R175H and R273H p53 mutants, commonly found in human cancer. These also led to decreased abundance of Lats2 and YAP phosphorylation through their dominant negative action (Fig. 6d). Similarly to p53 knockdown or HPV-E6, p53 mutants abolished the DVL repression of TEAD reporter activity (Fig. 6e). In the p53 mutant context, DVL consistently increased the TCF/LEF transcriptional activity of canonical Wnt (Fig. 6f). Because p53's most important function is to act as a transcription factor[42,43], we next used an artificial deletion mutant of the N-terminus transactivation domain (ΔTAD) of p53 to examine the intersecting role of DVL on YAP and Wnt[41,44]. Indeed, ΔTAD of p53 largely abolished DVL suppression of nuclear YAP while preserving DVL's potential effect on the canonical Wnt activity (Supplementary Fig. 8b).

Therefore, our results indicate that loss of p53 relieves YAP restriction by DVL and allows co-activation of TEAD and canonical Wnt activities by YAP and DVL, respectively. Examining TEAD and TCF/LEF transcriptional activities with respect to p53 status, YAP and DVL reciprocally suppressed TCF/LEF and TEAD activity in wt p53 context while the YAP suppressor role of DVL3 was abolished under p53-loss context, resulting in co-activation of TEAD and TCF/LEF activities by YAP and DVL with a p53-loss background (Fig. 6g). YAP inhibited tumor growth in vivo with suppression of Wnt activity[13]. We, therefore, examined the functional relevance of co-activation of those oncogenic transcriptional activities by YAP and DVL in terms of p53 context with xenograft experiments. Indeed, YAP in combination with DVL3 significantly increased the tumorigenic potential of 293 cells in a p53-loss context compared to wt p53 background (Fig. 6h). These results indicate that tumor suppressor p53 provides an important context of antagonistic interaction between nuclear YAP and the canonical Wnt pathway, and that the loss-of p53 in human cancer relieves the YAP restriction allowing co-activation of those oncogenic pathways by DVL. To further determine the effect of p53 on the co-activation of YAP and the Wnt pathway in clinical samples, we analyzed RNA expression data from primary human breast

cancer (1093 samples from TCGA) and chose CTGF and Axin2 transcripts, which are representative target genes of YAP and TCF/LEF transcriptional machinery, respectively. We grouped the patient samples by p53 status and dichotomized them according to CTGF and Axin2 transcript abundance (Supplementary Fig. 8c). When we analyzed the co-activation of YAP and the Wnt pathway in a patient cohort with respect to 20 years long-term survival, we found that increased abundance of CTGF and Axin2 was associated with worse prognosis in mutant p53 patients while the association was inverted in the wt p53 group (Fig. 6i), indicating the importance of p53 tumor suppressor context in co-activation of YAP and the canonical Wnt pathway in human cancer.

**Role of DVL in metabolic contexts.** The LKB1 (also known as STK11, serine/threonine kinase 11) is the key upstream activator of AMPK[45]. The mutational inactivation of LKB1 is a well-known genetic background of Peutz-Jeghers syndrome and is frequently found in various human cancers[46,47]. The LKB1/AMPK axis is critical to cellular energy homeostasis and epithelial polarity. Recent findings reveal that metabolic stress increases catalytic active AMPKα and Lats1, resulting in YAP phosphorylation and its subsequent cytosolic translocation[7,8]. Given the effect of YAP phosphorylation on DVL's dynamics, we extended our inquiry to metabolic regulation and the effect of DVL on YAP trafficking. We induced metabolic stress by treatment of 2-deoxy-glucose (2DG) and metformin (Met) in MCF-10A cells, then examined

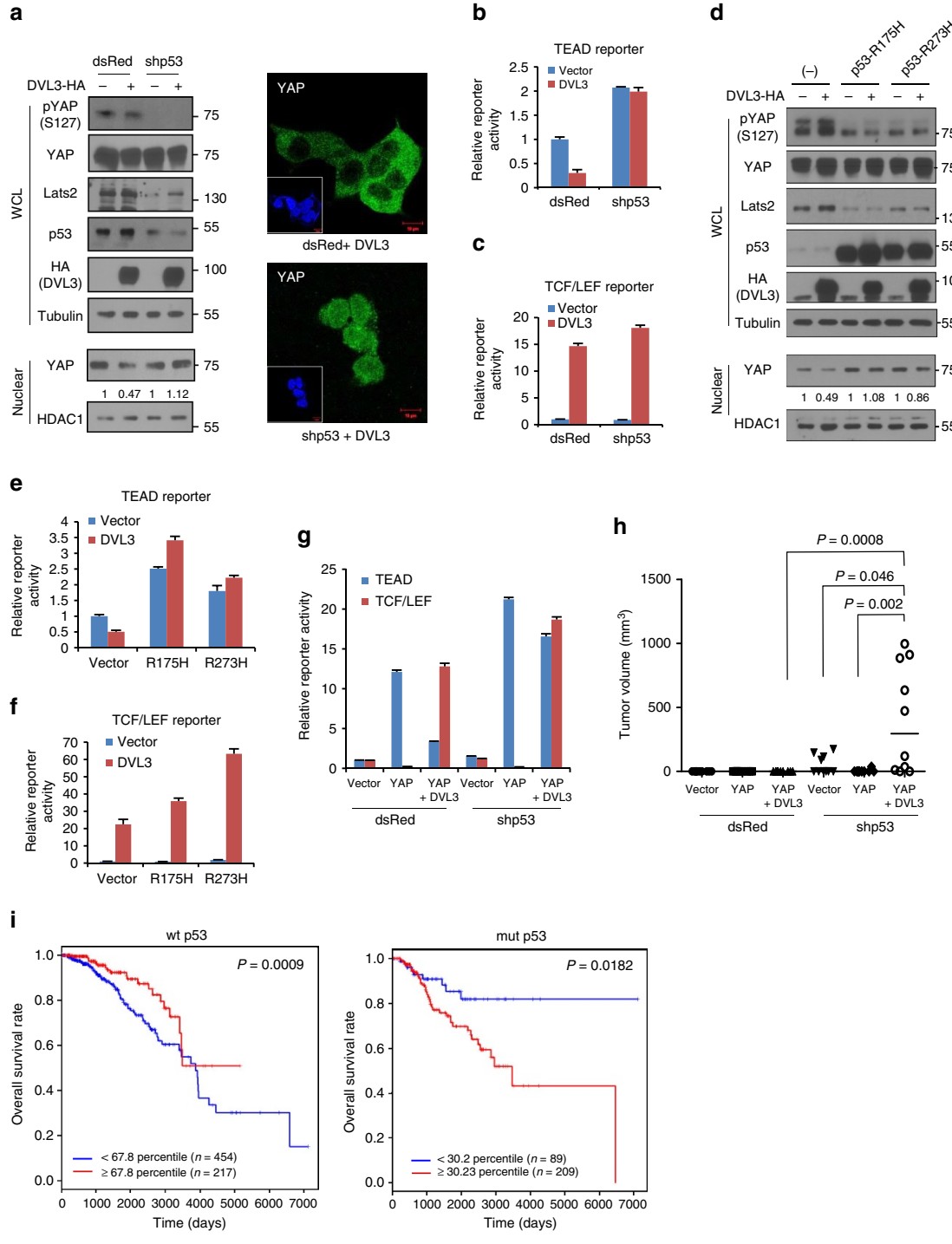

the YAP localization with an inducible DVL3 knockdown system. YAP was mainly localized in nuclear space under sparse culture condition and was translocated into the cytoplasm by metabolic stress (Fig. 7a). Importantly, YAP was retained in nuclear following inducible knockdown of endogenous DVL3 under metabolic stress. The role of metabolic stress on YAP cytoplasmic translocation was also defective in DVL-TKO cells (Fig. 7b), indicating that DVL is also critically required for YAP translocation induced by metabolic stress. Given the effect of LKB1/AMPK activity on DVL interaction with YAP, we next used dominant negative LKB1 (LKB1-KD) or AMPK (AMPK-KD) to examine the role of DVL in nuclear YAP activity in cells. Consistent with a previous observation[7], LKB1-KD or AMPK-KD increased TEAD reporter activity (Supplementary Fig. 9a). Similarly to p53 tumor suppressor context, LKB1-KD or AMPK-KD decreased YAP phosphorylation and subsequently abolished DVL's ability to suppress nuclear YAP abundance and TEAD transcriptional activity (Fig. 7c and Supplementary Fig. 9b). The activator function of DVL on canonical TCF/LEF transcriptional activity was maintained in the LKB1/AMPK mutant context, as seen in the co-activation of TEAD and TCF/LEF transcriptional activities. To further confirm the role of AMPK in DVL-mediated YAP translocation, we compared AMPKα wt and AMPKα1/α2 double-knockout (DKO) MEFs that lack the two AMPK catalytic subunits. Indeed, induction of DVL3 failed to translocate YAP into cytoplasm in the AMPK DKO MEFs (Fig. 7d), indicating the critical role of AMPKα in YAP cytoplasmic trafficking by DVL. To examine the role of endogenous LKB1 in YAP subcellular trafficking, we chose naturally LKB1-deficient (but having wt p53 and E-cadherin positive) A549 non-small-cell lung cancer cells and examined the YAP localization[44,48]. Indeed, YAP was diffusely localized at nuclear and cytoplasmic space regardless of cell–cell contact in epithelial A549 cells (Supplementary Fig. 9c). Importantly, overexpression of DVL3 was inactive in terms of YAP nuclear export in A549 cells (Fig. 7e), indicating that loss of LKB1 tumor suppressor abolishes DVL's YAP nuclear export function. The tumor suppressive role of LKB1 has been clearly demonstrated by experiments in which conditional LKB1 inactivation led to neoplastic transformation in a number of tissues[49]. To prove the functional relevance of LKB1 mutation context allowing co-activation of Wnt and nuclear YAP, we next performed an in vivo xenograft tumorigenic assay. Indeed, loss of LKB1 context increased tumorigenic potential by YAP and DVL in vivo as in the p53 tumor suppressor context (Fig. 7f).

Taken together, our results demonstrate that the intact tumor suppressor function of p53/Lats2 or LKB1/AMPK axes is required for inhibition of nuclear YAP by DVL. Conversely, mutational inactivation of p53/Lats2 or LKB1/AMPK axes in human cancer critically allows DVL to become a potent activator of the canonical Wnt pathway without suppression of nuclear YAP (Supplementary Fig. 9d).

## Discussion

The Hippo/YAP and canonical Wnt pathways are highly conserved signaling cascades regulating cell-to-cell interaction and contact inhibition, and hyperactive nuclear YAP and Wnt activity are bona fide oncogenes in many types of human cancer[2,4]. Moreover, the canonical Wnt and Hippo pathways are closely connected to each other[6,11]. Recent studies have focused on a molecular mechanism whereby YAP represses the Wnt activity via interaction with β-catenin, DVL2 and the Axin-GSK-3 complex[13,14,16]. These observations raise the paradox of a tumor suppressive function of cytoplasmic YAP[2,15]. In this study, we show that DVL, a scaffolding protein of the canonical Wnt pathway as well as an integral member of non-canonical planar cell polarity (PCP) signaling, binds to phosphorylated YAP with the PDZ domain and PY motif, subsequently regulating nuclear-cytoplasmic trafficking of YAP.

Note that other scaffolding proteins of the Wnt pathway have a similar function. The APC (adenomatous polyposis coli) harbors highly conserved NES, whose mutational inactivation in cancer cells results in nuclear accumulation of β-catenin[29,30]. Intriguingly, loss of APC activates YAP by interacting with Lats1 in a GSK-3 dependent manner[50]. Further, Axin regulates the Snail-mediated epithelial-mesenchymal transition (EMT) process by acting as a nuclear exporter of GSK-3[31,51]. The GSK-3 shuttling function of Axin is also required for phosphorylation of the membranous LRP6 Wnt co-receptor and subsequent activation of the intracellular Wnt activity[52]. Interestingly, we found that soluble Wnt ligands activate YAP and subsequently decrease affinity to DVL and translocation into nuclear space. Considering that DVL is directly regulated by Frizzled (FZD) receptors, our observations provide an interesting mechanistic insight into intracellular dynamics between Wnt and Hippo.

As a key scaffolding protein of the Wnt pathway, DVL relays extracellular signals from FZD receptors to downstream effectors, interacts with a wide range of proteins, and is involved in diverse signaling pathways including canonical Wnt and non-canonical PCP[21,23]. While the phosphorylation-dependent YAP translocation by activation of the Hippo pathway and AMPK is well known[5–8], a molecular effector regulating the YAP nuclear-cytoplasmic shuttle has not yet been identified. We show that NES embedded in DVL is responsible for cytoplasmic translocation of phosphorylated YAP. Importantly, DVL is required for nuclear-cytoplasmic trafficking of YAP induced by contact inhibition, α-catenin, E-cadherin, and metabolic stress, indicating that

**Fig. 6** Loss of p53/Lats2 tumor suppressor allows co-activation of YAP and canonical Wnt activity by DVL. **a** The 293 cells were transfected with control (dsRed) or shRNA for p53 (dsRed-shp53) in combination with HA-tagged DVL3 expression vector. Whole-cell lysates (WCL) were immunoblotted with indicated antibodies, and nuclear fractions were used for nuclear YAP abundance (left panels). Endogenous YAP localization was determined by confocal microscopy (right panels). Inset, DAPI nuclear stain; Scale bar, 10 μm. **b**, **c** The TEAD (**b**) or TCF/LEF (**c**) reporter constructs were co-transfected with YAP and DVL3 in control transfected cell (dsRed) or shRNA for p53 transfected cells, and the relative reporter activity was measured from triplicate experiments (mean ± SD). **d** The 293 cells were transfected with control (−) or mutants p53 (p53-R175H, p53-R273H) in combination with HA-tagged DVL3 expression vector. Whole-cell lysates (WCL) were immunoblotted, and nuclear fractions were used for nuclear YAP abundance. **e**, **f** The TEAD (**e**) and TCF/LEF (**f**) reporter construct was co-transfected with DVL in empty vector transfected cell (−) or p53 mutants transfected cells, and the relative reporter activity was measured from triplicate experiments (mean ± SD). **g** The TEAD or TCF/LEF reporter constructs were co-transfected with YAP and DVL as indicated in control transfected cell (dsRed) or shRNA for p53 transfected cells. The relative reporter activity was measured from triplicate experiments. Data presented as mean ± SD. **h** The 293 cells ($1 \times 10^6$) were transiently transfected with YAP and DVL3 as indicated in combination with dsRed control or shRNA for p53, and the cells were inoculated into the flank of athymic nude mice ($n = 10$). Tumor volume was measured 5 weeks post-injection. Statistical significance was determined by Mann–Whitney test. **i** Kaplan–Meier survival graphs for breast cancer patients with wt or mutant p53 status on the basis of CTGF and Axin2 transcript abundances at an optimal threshold indicated by percentile numbers. Samples with high abundance of CTGF and Axin2 are represented with red lines. See Supplementary Fig. S8c for scatter plot of CTGF and Axin2 transcript abundance. A log-rank test was used to calculate statistical significances

the intracellular shuttling function of DVL is essential to various developmental and oncogenic contexts. Previous observation showed that knockdown of YAP resulted in nuclear accumulation of DVL[13], suggesting that the nuclear export machinery of DVL and YAP are mutually dependent. It should be noted that 14-3-3 is required for cytoplasmic retention of phosphorylated YAP[5], and that we found DVL interacts with 14-3-3. Although the key role of 14-3-3 in nuclear-cytoplasmic retention of phosphorylated nuclear proteins such as cdc-25, FKHRL1, and p65-IκBα is well known[53–55], 14-3-3 itself does not have a nuclear export function[54]. These results, together with our observations, suggest that 14-3-3 may facilitate DVL's nuclear export of phosphorylated YAP. Further study is required to delineate the role of 14-3-3 in

DVL-mediated nuclear-cytoplasmic dynamics of YAP and other phosphorylated transcriptional machinery.

The phosphorylation of YAP is important not only for its intracellular localization but also for protein stability and binding to other partners[2]. Although DVL binds and exports phosphorylated YAP, increased nuclear YAP by loss of DVL remained unphosphorylated. Interestingly, Ser127 phosphorylation is important for interaction with DVL; however, Ser127 of nuclear YAP is still unphosphorylated in ASA mutant expressing cells, suggesting that Ser127 phosphorylation may be reciprocally coupled to the nuclear export function of DVL. While this study mainly focused on nuclear-cytoplasmic trafficking functions of DVL, the role of DVL in YAP phosphorylation together with the

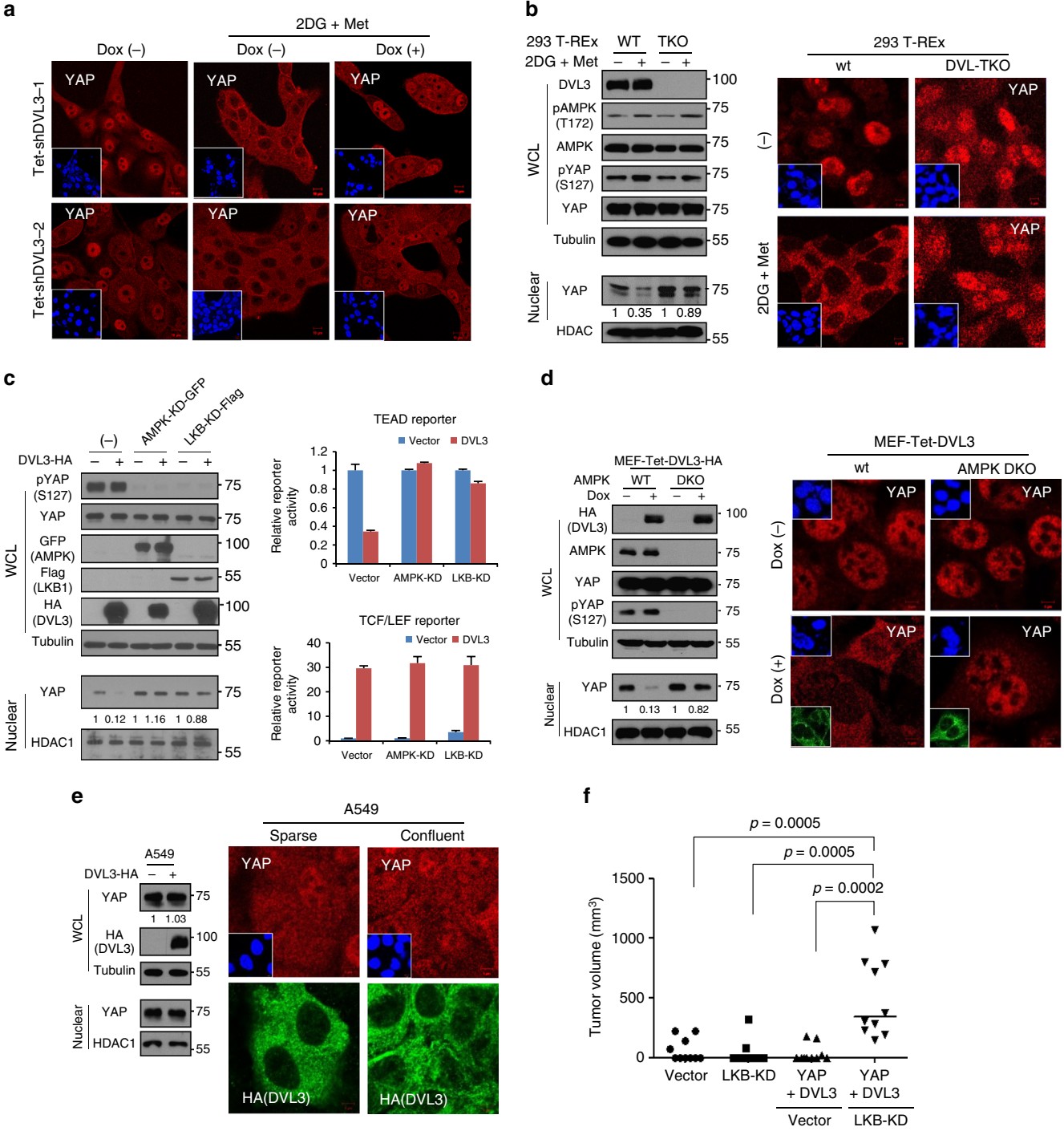

Lats kinase complex in nuclear space (such as phosphorylation at Ser109 and/or Ser381) needs further study to understand in-depth molecular dynamics in the Hippo and Wnt pathways. It should be noted that the modes of interaction and intracellular function of DVL closely resemble Angiomotin (Amot) inhibiting YAP activity[26]. While the Amot was identified as an angiostatin-binding protein involved in endothelial polarization and migration, recent studies have revealed that Amot is an important regulator of intracellular localization and nuclear YAP activity[26,56,57]. Amot also harbors a conserved PY motif and C-terminus PDZ-binding motif as do DVL and YAP, respectively[26,58]. Interestingly, Amot is also phosphorylated by Lats kinase and subsequent interaction with 14-3-3[56,58]. Similarly to DVL, the Amot family binds and sequesters phosphorylated YAP[2,26,56], while the nuclear export function of Amot has not yet been determined.

PCP signaling controls polarization of cells in an epithelial sheet via an FZD–DVL complex mediated by non-canonical Wnt signaling. In this process, DVL (*Dsh* in *Drosophila*) and FZD play a central role in PCP signaling, asymmetric relocalization of Dsh and cytoskeletal reorganization of epithelial cells[59]. Although many recent findings suggest that the Hippo pathway is tightly connected with PCP in development and epithelial polarity in mammal[3,60], the molecular dynamics are largely unknown. Our observations thus provide an interesting molecular link between Hippo and PCP regulation in development and epithelial polarity in human cancer. It should be noted that we could not find typical NES in *Drosophila* Dsh while there is highly conserved NES at the distal region of the DEP domain in *Ascidian* and *Xenopus*[23]. Further study is needed to identify the evolutionary role of DVL in epithelial polarity.

The LKB1 directly regulates a number of conserved targets, the most important being AMPK, a major sensor of cellular metabolic stress[45]. Studies of the LKB1/AMPK axis have discovered a novel signaling-pathway that links it with metabolism and epithelial polarity. For example, activation of LKB1 by STRAD rapidly leads to complete epithelial polarity accompanied by remodeling of the actin cytoskeleton[61], and constitutive activation of the LKB1/AMPK axis is required for epithelial polarity and tight junction formation regardless of cell–cell contact[62–65]. Note that E-cadherin regulates AMPK activity by recruiting the LKB1/STRAD complex at the adherens junction[65,66] and that the E-cadherin/α-catenin complex functions as a strong upstream regulator of the Hippo pathway and subcellular localization of YAP[36–38]. Therefore, the molecular mechanism by which LKB1/AMPK-dependent DVL controls epithelial polarity proteins such as PAR, Crumbs and Scribble in development and human cancer requires further investigation.

The tumor suppressor p53 is most frequently inactivated in human cancer, its transcriptional activity playing key roles in tumor suppression[42,43]. The transcriptional function of p53 also suppresses canonical Wnt activity and Snail-mediated EMT by targeting the untranslated regions of a set of genes encoding key elements of the Wnt pathway[41,44]. We found that the transcriptional function of p53 is also critically important in DVL's function on YAP trafficking. Notably, p53 and Lats2 tumor suppressors have been linked in a positive feed-forward loop by which Lats2 strengthens p53 function, and p53 upregulates Lats2 on the transcriptional level[40]. Lats2 is also suppressed by an epigenetic mechanism independent of p53 mutational status in many types of human cancers[67]. The role of the LKB1/AMPK axis on DVL's nuclear export of YAP is especially interesting because the LKB1 tumor suppressor is mainly associated with metabolic aspects of human cancer. Thus, our observations provide a novel molecular mechanistic insight into the reciprocal restriction between Wnt and YAP in a tumor suppressor context-dependent manner.

## Methods

**Cell culture and immunoblot analysis.** MCF-7, SK-BR-3, HCT-116, SW480, A549, and 293 cells obtained from ATCC were routinely cultured in Dulbecco's Modified Eagle's Medium (DMEM) medium containing 10% fetal bovine serum (FBS). MDA-MB-231 cells (a gift from G. Mills) were cultured in a RPMI1640 with 5% FBS. MCF-10A cells (a gift from M. Wicha) were cultured in DMEM/F12 with 5% horse serum, 20 μg ml$^{-1}$ EGF, 0.5 μg ml$^{-1}$ hydrocortisone, 0.1 μg ml$^{-1}$ cholera toxin, 5 μg ml$^{-1}$ insulin and 100 IU ml$^{-1}$ penicillin/streptomycin. AMPKα1/ α2 double-knockout MEF, Lats1/2 double-knockout MEF and 293A cells were kindly provided by H. W. Park (Yonsei University, Seoul, Korea) and cultured in DMEM medium. The wt and DVL1/2/3-triple knockout 293 T-REx cells were described previously[22]. Mycoplasma infection was tested regularly with a PCR-based kit (MP0040, Sigma). Cell lines were authenticated as described recently[68]. The transfection was performed by Lipofectamine 2000 according to the manufacturer's protocol (Invitrogen). For the western blot analyses, protein extracts were prepared in Triton X-100 lysis buffer. The nuclear protein abundances of YAP were determined from nuclear-cytosolic fractionation of protein lysates with hypotonic buffer as described previously[31,69]. Briefly, the cells (1 × 10$^6$) were collected into micro-centrifuge tubes and treated with 400 μl of hypotonic buffer (10 mM HEPES, pH 7.9; 10 mM KCl; 1 mM dithiothreitol (DTT) with protease inhibitors) on ice for 5 min. The cell membrane was ruptured by adding 10% NP-40 to a final concentration of 0.6%, then vigorously vortexed for 10 s followed by high-speed centrifuge for 30 s. The supernatant cytosolic fractions were collected separately, and nuclear pellets were washed with ice-cold PBS twice. Nuclear protein was extracted with hypertonic buffer (20 mM HEPES, pH 7.9; 0.4 M NaCl; 1 mM DTT with protease inhibitors) on ice for 15 min followed by high-speed centrifuge. Relative nuclear YAP abundance compared to loading control HDAC1 was determined by the ImageJ program downloaded from NIH (https://imagej.nih.gov/ij/). Wnt1 extract and Wnt3a conditioned were prepared from RAC311-Wnt1 cells and L-Wnt3a cell as described previously[51]. Antibodies against YAP (sc-101199, Santa Cruz, 1:1,000, 1:200 for IF), phospho-S127 YAP (4911S, Cell Signaling Technology, 1:1000), DVL1 (sc-8025, Santa Cruz, 1:1000), DVL2 (sc-8026, Santa Cruz, 1:1000), DVL3 (sc-8027, Santa Cruz, 1:1000, 1:100 for IF), α-catenin (sc-7894, Santa Cruz, 1:1000), Lats2 (ab70565, Abcam, 1:1000), LKB1 (sc-32245, Santa

**Fig. 7** Role of LKB1/AMPK tumor suppressor axis on DVL's function on YAP. **a** The MCF-10A cells expressing tetracycline-inducible shRNA against DVL3 were cultured under sparse culture condition and treated with 2-deoxyglucose (2DG, 3 mM) and metformin (Met, 5 mM) for 16 h period in absence (−) or presence (+) of doxycycline (Dox). Endogenous YAP localization was determined by confocal microscopy. Inset, DAPI nuclear stain; Scale bar, 10 μm. **b** The wt and DVL-TKO 293 T-REx cells were treated with 2DG and Met, and nuclear YAP abundance was then determined by immunoblot analysis (left panels) and confocal microscopy (right panels). Inset, DAPI nuclear stain; Scale bar, 5 μm. **c** Kinase-dead GFP-fused AMPK or flag-tagged LKB1 mutant was co-transfected with control (−) or HA-tagged DVL3 (+) in 293 cells. Whole-cell lysates (WCL) were immunoblotted with indicated antibodies, and nuclear fractions were used for nuclear YAP abundance (left panels). The TEAD (right upper) or TCF/LEF (right lower) reporter was co-transfected with DVL in empty vector transfected cell (vector) or kinase-dead AMPK (AMPK-KD) or LKB mutant (LKB-KD), and the relative fold repression of reporter activity was measured from triplicate experiments (mean ± SD). **d** The wt and AMPKα1/α2 double-knockout (DKO) MEFs were stably transfected with inducible DVL3 and nuclear YAP abundance was examined by immunoblot analysis (left panels) and immunofluorescence (right panels) in the absence (−) or presence (+) of doxycycline for a 48 h period. Upper inset, DAPI nuclear stain; Lower inset, HA (DVL3); Scale bar, 5 μm. **e** The A549 cells were stably transfected with inducible DVL3, and nuclear YAP abundance and DVL3 localization under sparse or confluent culture condition were examined by immunoblot analysis (left panels) and immunofluorescence (right panels) in presence of doxycycline. Inset, DAPI nuclear stain; Scale bar, 5 μm. **f** The 293 cells (1 × 10$^6$) were transiently transfected with YAP and DVL3 in combination with vector control or LKB-KD mutant, and the cells were inoculated into the flank of athymic nude mice (*n* = 10). The empty vector or LKB-KD transfected cells served as control. Tumor volume was measured 5 weeks end-point (Mann–Whitney test)

Cruz, 1:1000), AMPK (2793S, Cell Signaling Technology, 1:1000), phospho-AMPK (2535S, Cell Signaling Technology, 1:1000), HA (901501, Bio Legend, 1:2500, 1:200 for IF), p53 (sc-126, Santa Cruz, 1:1000), E-cadherin (HECD, AR17-MA0001, AbFrontier, 1:5000, 1:500 for IF), HDAC1 (sc-7872, Santa Cruz, 1:1000), GFP (GF-PA0043, AbFrontier, 1:1000), Flag (F-3165, Sigma, 1:5000), pan-TEAD (13295, Cell Signaling Technology, 1:2000, 1:100 for IF) and Tubulin (LF-PA0146, AbFrontier, 1:2500) were obtained from the commercial vendors. Phos-tag gel was purchased from WAKO chemicals (AAL-107).

**Plasmids and RNA-mediated interference**. HA-tagged DVL1, DVL2, and DVL3 expression vectors were kindly provided by E. Fearon (University of Michigan). Deletion or point mutants of DVL3 (ΔDIX, ΔPDZ, ΔDEP, NES-ASA, and ΔPDZ/ΔPY) and YAP deletion mutants of PDZ-binding domain, WW domain and K494R were generated by a PCR-based method. The expression vector pCMV-flag-YAP (plasmid number 19045), pCMV-flag-S127A YAP (plasmid number 19050), pCMV-flag-5SA-YAP (plasmid number 27371), 3xflag-pCMV5-TAZ (plasmid number 24809), pLKO1-shYAP (plasmid number 27368 and 27369), pcDNA-HA-Lats2-K655R (kinase-dead, plasmid number 33100), pAMPK alpha2 K45R (kinase-dead, plasmid number 15992), pBabe-LKB1-K78I (kinase-dead, plasmid number 8593), 14-3-3 gamma (plasmid number 52535) and pcDNA3-alpha-catenin (plasmid number 24194) were obtained from Addgene. Reporter construct having 8 × wt TEAD-binding sites (plasmid number 83467), 7 × mutant TEAD-binding sites (plasmid number 83466), 8 × TCF/LEF-binding sites (plasmid number 12456), and mutant-binding sites (plasmid number 12457) were obtained from Addgene. The Tet-pLKO-puro vector (plasmid number 21915, Addgene) was used for inducible shRNA knockdown. The target sequences of shRNA were 5′-cgacc-cagctataagttcttcttca for human DVL3-1 and 5′-caatgacacagagacggactctttg for human shDVL3-2. The expression vectors for shRNA against p53 and expression vectors for HPV-E6 and mutant p53 were as described previously[41]. Tetracycline-inducible wt or NES-mutant DVL3 expression vector was generated with the pTRIPZ lentiviral system (Open Biosystems) by replacing RFP.

**Immunoprecipitation and immunofluorescence**. For immunoprecipitation analysis, whole-cell Triton X-100 lysates were incubated with Flag-M2 agarose (Sigma) or Ni-NTA beads (Invitrogen) and washed with lysis buffer three times. The recovered proteins were resolved by SDS-PAGE and subjected for immunoblot analysis. For immunofluorescence study, the cells were washed twice with ice-cold PBS and incubated for 15 min at room temperature with 3% formaldehyde in PBS. The cells were permeabilized with 0.5% Triton X-100 for 5 min and then blocked for 1 h in PBS containing 3% bovine serum albumin followed by incubation with primary antibody overnight at 4 °C. Cells were then washed three times with PBS containing 0.1% Tween 20 followed by incubation with anti-mouse-Alexa Fluor-488 (for green) or anti-rabbit-Alexa Fluor-594 (for red) secondary antibody. Cellular fluorescence was monitored using confocal microscopy (Zeiss LSM780). Functional blocking of endogenous E-cadherin complex was performed by directly adding an HECD antibody (10 μg/ml) to normal culture medium for 16 h prior to immunofluorescence study of YAP.

**Quantitative reverse transcription PCR (RT-PCR) and reporter assay**. Total RNA was isolated using TRIzol reagent (Invitrogen) following the manufacturer's protocol. The SuperScript III synthesis kit (Invitrogen) was used to generate complementary DNA. Real-time quantitative PCR analysis for CTGF transcripts was performed with an ABI-7300 instrument under standard conditions and SBGR mix ($n = 3$). The expression of ΔCt value from each sample was calculated by normalizing with GAPDH. Primer specificity and PCR process were verified by dissociation curve after PCR reaction. The primer sequences for qPCR were 5′-accagctcctcctcactaacc for DVL1 forward (F), 5′-tcatgtcactcttcaccgtca for DVL1 reverse (R), 5′-catgagaatctggagcctgag for DVL2-F, 5′-atgctcactgctgtctctcct for DVL2-R, 5′-agaaggtttctcggattgagc for DVL3-F, 5′-tgttgagagtgaccgtgatga for DVL3-R, 5′-caaaatctccaagcctatcaagtt for CTGF-F, 5′-actccacagaatttagctcggtat for CTGF-R, 5′-atgggtgtgaaccatgagaag for GAPDH-F, and 5′-agttgtcatggatgaccttgg for GAPDH-R. For TEAD or TCF/LEF reporter assay, the cells were transfected with 50 ng of the reporter vectors and 1 ng of pSV-Renilla expression vector in combination with YAP and/or DVL3 as indicated. Luciferase and *renilla* activities were measured using the dual-luciferase reporter system kit (Promega), and the luciferase activity was normalized with *renilla* activity. The results are expressed as the averages of the ratios of the reporter activities from triplicate experiments.

**Soft agar assay**. For anchorage-independent soft agar assay, cells stably transfected with pLKO-tet-shDVL3 were suspended at $1 \times 10^4$ cells per 6-well plate with 1 ml of 0.3% low-melting agar in 2 × DMEM containing 20% FBS and overlaid above a layer of 1 ml of 1% agar in the same medium. After 2 weeks incubation with or without doxycycline as indicated, colonies were visualized by staining with 0.05% crystal violet in 10% ethanol for 30 min and viable colonies that contained > 50 cells were counted from five fields with a stereomicroscope. Representative colonies were photographed and two independent experiments were performed.

**Gene expression analysis of clinical samples**. Publicly available mRNASeq data of 1098 samples of breast cancer (BRCA) including long-term survival information

from The Cancer Genome Atlas (TCGA) was downloaded (https://gdac.broadinstitute.org). The illuminahiseq_rnaseqv2-RSEM_genes_normalized (MD5) was log2 transformed and the relative transcript abundance of DVL homologs was compared using one-way ANOVA and Tukey's HSD test. Mutational status of p53 was obtained from Mutation Annotation Format (MAF) from the Firehose website. To generate Kaplan–Meier plots, clinical samples were grouped by p53 status and dichotomized by Axin2 and CTGF transcript abundance. The scatter plots of CTGF and Axin2 abundances were obtained using R. The high and low abundance subsets were determined based on the median transcript abundance of CTGF and Axin2, yielding groups with the most significant differences in 20-year survival based on the log-rank test. The Kaplan–Meier plots were then generated for the respective groups using the R package survival.

**In vivo xenograft assay**. All animal experiments were performed in accordance with guidelines of the Institutional Animal Care and Use Committee of Yonsei University and approved by the Animal Care Committee of the Yonsei University College of Dentistry and National Cancer Center Research Institute. Female athymic nude mice (6-weeks-old) were used for xenograft assays into flank subcutaneous tissue. The 293 cells were transiently transfected with expression vectors of DVL3 and YAP in combination with shp53 or LKB-KD as indicated prior to 48 h in vivo inoculation. The cells ($1 \times 10^6$) were resuspended in 100 μl of PBS and injected into flank subcutaneous tissue. The tumorigenic capacity was measured twice a week using a digital caliper. The mice were sacrificed after 5 weeks endpoint, and the tumor volume was calculated using the equation $V$ (in mm³) = $(a \times b^2)/2$, where $a$ is the longest and $b$ the shortest diameter.

**Statistical analysis**. All statistical analysis of reporter assay, RT-PCR, and soft agar assay was performed with two-tailed Student's $t$-tests; data are expressed as means and s.d. The double asterisks denote $p < 0.01$, one asterisk denoting $p < 0.05$. Statistical significance of animal experiments was determined using the Mann–Whitney test. No statistical method was used to predetermine sample size.

**Data availability**. The data that support the findings of this study are available from the corresponding author upon reasonable request.

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

## Acknowledgements

We thank E. Tunkle for preparation of the manuscript and K.Y. Kim for statistical analysis. We thank professor H.W. Park at Yonsei University for generously providing the Last1/2$^{-/-}$ 293 A, Last1/2$^{-/-}$ MEF and AMPK-null MEF cells. We thank Yonsei Advanced Imaging Center in cooperation with Carl Zeiss Microscopy, Yonsei University College of Medicine, for technical assistance. This work was supported by grants from the National Research Foundation of Korea (NFR-2017R1A2B3002241, NRF-2016R1E1A1A01942724, NRF-2017R1C1B1012464, NRF-2018M3A9E2022820) funded by the Korean government (MSIP), a grant from the National Research Foundation of Korea (NRF-2014R1A6A3A04055110) and a grant from the Korea Health Industry Development Institute (KHIDI) funded by the Ministry for Health & Welfare Korea (HI17C2586). V.B. was supported by the Czech Science Foundation (GA17-16680S,

GA18-17658S), by the Neuron Fund for Support of Science and Marie Curie ITN WntsApp (Grant no. 608180). Petra Paclíková is a Brno Ph.D. Talent Scholarship Holder, funded by the Brno City Municipality.

## Author contributions

Y.L. and N.H.K. performed all experiments; E.S.C., J.H.Y., Y.H.C., H.E.K., J.S.Y., Y.S.Y., and S.Y.C. supported in vitro and in vivo experiments; S.B.C. performed bioinformatics analysis; C.G.K., S.-H.L. and S.-Y.K. performed in vivo experiments; P.P., T.W.R., and V.B. generated and performed DVL-TKO cell experiments; N.H.K., H.S.K., and J.I.Y. planned all experiments, analyzed the data, and wrote the manuscript.

## Additional information

**Competing interests:** The authors declare no competing interests.

