## [Peer Review File · Nature Communications]

Reviewers' comments:

Reviewer #1 (Remarks to the Author):

Previous studies have established a functional interaction between Wnt and Hippo although the precise relationship is not clear. In addition, the Hippo pathway component YAP has been reported to interact with Dvl, leading to cytoplasmic localization and inhibition of Dvl. In this study, the authors reexamined the relationship between Dvl and YAP. They confirmed the interaction between Dvl and YAP/TAZ. Overexpression of Dvl3 increased YAP cytoplasmic localization while knockdown of Dvl3 increased nuclear YAP. The authors proposed a model that Dvl is responsible for cytoplasmic localization of the phosphorylated YAP by binding to the phosphorylated YAP to promote YAP nuclear export. Moreover, Dvl is proposed as a tumor suppressor by inhibiting YAP.

A clear understanding about the YAP cytoplasmic localization will have some significance to our understanding of the Hippo pathway. Furthermore, elucidation of the relationship between Dvl and YAP (whether YAP inhibits Dvl or vice versa) will help to clarify some confusion about the relation between Wnt and Hippo. However, the current study has not provided convincing data to address these key issues. In addition, there is little evidence in the literature to support Dvl as a tumor suppressor while many studies would support Dvl as an oncogene to promote Wnt signaling.

It is important to demonstrate that Dvl selectively binds the phosphorylated YAP because phosphorylation is known to play a major role in YAP regulation and the effect of Dvl on YAP appears to be dependent on YAP phosphorylation. A simple in vitro protein-protein interaction will convincingly demonstrate whether Dvl3 binds to phosphorylated or unphosphorylated YAP. The observed increase of Dvl-YAP interaction presented in the manuscript can be easily explained as a consequence of YAP cytoplasmic localization that is caused by YAP phosphorylation, rather than the enhanced affinity of Dvl towards the phosphorylated YAP.

It has been reported by several groups that TEAD functions to retain YAP in the nucleus whereas 14-3-3 functions to retain phosphorylated YAP in the cytoplasm. What is the relative importance of Dvl in YAP subcellular localization when comparing to TEAD and 14-3-3? To really demonstrate a critical role of Dvl in YAP localization, the authors should generate Dvl3 knockout cells, then determine YAP localization under various conditions, such as cell density and energy stress, to unambiguously show that Dvl is indeed important for YAP nuclear export.

Fig.S1b-c. Because different antibodies were used to detect different Dvl proteins, simply based on the Western intensity the authors cannot conclude that Dvl3 is most abundant in these cancer cells.

Fig.1a. A control, such as DVL3 knockdown or knockout, is needed to show that the Dvl3 antibody is specific in the staining.

Fig.2c. Cell density has a very important effect on YAP localization. Based on the DNA staining (insert), it appears that the Dox+ cells have lower cell density, which should have more nuclear YAP. It is extremely important to have exactly same cell density in order to look at YAP localization. Does Dvl3 knockdown affect YAP phosphorylation?

Fig.S4a. The cytoplasmic YAP levels and expression of Dvl3 should be included. It was reported that the PDZ domain in Dvl is required for interaction with TAZ (Dev. Cell. 18, 579-591, 2010). Do the authors have explanation why deletion of YAP PDZ domain has no effect on the TEAD reporter?

Fig.3c. The experimental conditions could be problematic. For the un-transfected cells (no red color), YAP was cytoplasmic in the Dvl3-wt panel but more nuclear YAP in the Dvl3-ASA-mut panel. Therefore, the difference could largely be due to the difference in culture conditions, which

would affect YAP localization. Can Dvl3 expression induce YAP cytoplasmic localization in LATS1 and LATS2 knockout cells?

Fig.5a. Key controls of YAP localization without transfection of E-cadherin or α -catenin need to be included. 5b. YAP levels in the cytoplasmic fraction should be included. 5c. Is the color for HECD antibody (red, indicated by arrows) same as YAP (red)?

Fig.7a-c. Both AMPK and LKB1 knockout cells are widely available. Why the authors used overexpression of dominant negative mutants to do these experiments instead of the much cleaner experiments with knockout cells? The data in 7a is strange. Overexpression of AMPK-KD or LKB1-KD caused a dramatic dephosphorylation and increased nuclear localization, why the TEAD reporter activity was not increased by the dominant negative AMPK or LKB1?

7d. An essential control of LKB1-KD is missing. Moreover, this data is not sufficient to conclude that LKB1-KD function through YAP to promote tumor growth.

7e. This manuscript provides insufficient data for the model, which is rather speculative.

Page 11, line 236, typo "hemophilic"

Page 15, line 326, should be Fig.7d. Fig 7c is not cited in the text.

Reviewer #2 (Remarks to the Author):

Although the phosphorylation dependent cytoplasmic localization of YAP/TAZ has been widely studied, the key effector molecule(s) that determine the nucleocytoplasmic shuttling of YAP/TAZ have not been well investigated. This manuscript describes a regulation of YAP activity by DVL3, a known component of the Wnt signaling pathway. Cytoplasmic YAP/TAZ has been shown to restrict Wnt signaling by limiting the activity (Varelas, 2010) and nuclear localization of DVL (Barry, 2013). Reciprocal to these findings, in the current manuscript Lee et al studied the regulation of YAP activity by DVL. The authors show that DVL3 is the nucleocytoplasmic shuttling effector of phosphorylated YAP which is induced by upstream signaling such as E-cad, α -cat and AMPK via Lats. Depletion of DVL3 or mutation of the NES of DVL3 (DVL3-ASA) increases the nuclear level of YAP. This is an important new contribution to the field.

Authors also claim that the dysregulation of p53 or LKB1 signaling pathway in breast cancer relieves the restriction of YAP activity by DVL3, and leads to the co-activation of TEAD and TCF/LEF transcriptional activities. This suggests the tumor suppressor role of p53 and LKB1 signaling through the regulation of YAP activity by DVL3.

Overall, the manuscript is well written and the conclusion of this manuscript is supported by experimental results. However, the authors should provide more experimental results and/or discussion to support their proposed model.

1. In many places the authors suggest that YAP is regulated by the Wnt pathway, but this has not yet been shown. DVL is a component of the Wnt pathway, but it is possible that it has a distinct function in the Hippo pathway. This is an important question and the authors should test whether stimulation with a soluble Wnt affects YAP through this mechanism. It doesn't matter which result they get, but they should perform the experiments and discuss this issue.

2. Varelas et al (Dev cell 2010; 18(4) 579–591) identified the TAZ binding domain in DVL2. Rescue of DVL3 depleted cells with a mutant defective for YAP binding followed by immunofluorescence staining of YAP would add further support of DVL3 as a shuttling effector of YAP.

3. The reports by Mana-Capelli et al (Mol Biol Cell. 2014;25(10):1676-85) and Moleirinho et al (Mol Biol Cell. 2014;25(10):1676-85) should be mentioned and discussed as they showed the cytoplasmic retention of YAP by angiomotins.

4. The authors should show the phosphorylation status of nuclear YAP in DVL3 depleted cells (Fig 2b, 2c, Sup Fig 3) and DVL3-ASA mutant expressing cells (Fig 3d). The decrease in YAP phosphorylation by the inactivation of upstream kinases enriches the level of nuclear YAP protein. If DVL3 plays a role as a nucleocytoplasmic shuttling effector of YAP without regulating the upstream kinases of YAP, depletion of DVL3 or overexpression of DVL3-ASA should increase the level of phosphorylated YAP in the nucleus.

5. Fig 4b and 5b needs Western blot results of YAP in the whole cell lysates for the proper quantification.

6. Authors should include the immunofluorescence staining results of control 293 cells in Fig 5a.

There are numerous typos and errors throughout the manuscript:

1. DLV should be corrected to DVL (line #137, #313)

2. Misspelling of suppresses (line #122)

3. Fig. 7c at line 326 should refer to Fig. 7d.

4. Hemophilic should be corrected to homophilic (line #236, #775)

5. Typo of E-cadherin (line #238)

6. 293T should be changed to 293 (line #423, #532), as the entire manuscript described that authors used 293 cells.

7. There is a labeling error in Fig 2c.

Response to Reviewers (NCOMMS-17-16923-T)

We appreciate the reviewers' helpful comments and constructive criticisms on our manuscript. We have performed a large number of additional in vitro and in vivo experiments to address all the points raised by the reviewers. Our responses to the reviewers are provided in bold below.

Point-by-Point responses

Reviewer #1:

Previous studies have established a functional interaction between Wnt and Hippo although the precise relationship is not clear. In addition, the Hippo pathway component YAP has been reported to interact with Dvl, leading to cytoplasmic localization and inhibition of Dvl. In this study, the authors reexamined the relationship between Dvl and YAP. They confirmed the interaction between Dvl and YAP/TAZ. Overexpression of Dvl3 increased YAP cytoplasmic localization while knockdown of Dvl3 increased nuclear YAP. The authors proposed a model that Dvl is responsible for cytoplasmic localization of the phosphorylated YAP by binding to the phosphorylated YAP to promote YAP nuclear export. Moreover, Dvl is proposed as a tumor suppressor by inhibiting YAP.

A clear understanding about the YAP cytoplasmic localization will have some significance to our understanding of the Hippo pathway. Furthermore, elucidation of the relationship between Dvl and YAP (whether YAP inhibits Dvl or vice versa) will help to clarify some confusion about the relation between Wnt and Hippo. However, the current study has not provided convincing data to address these key issues. In addition, there is little evidence in the literature to support Dvl as a tumor suppressor while many studies would support Dvl as an oncogene to promote Wnt signaling.

Response) We appreciate the reviewer's helpful comment on DVL function intersecting the Wnt and Hippo pathways. To address this issue, we further performed experiments of YAP trafficking with soluble Wnt, an issue also raised by reviewer 2. Consistent with previous observations by Park HW et al (*Cell* 2015, 162, 780-794), soluble Wnt1 and Wnt3a ligands promote YAP nuclear localization via increased active YAP as determined by pS127-YAP antibody and mobility shift on a phos-tag gel analysis. In accordance with our proposed model in this study, molecular interaction between DVL and YAP was decreased by soluble Wnt ligands (Fig. 4f). In immunofluorescence study, Wnt ligandS induced nuclear localization of YAP although the DVL was mainly retained in cytoplasmic space (Fig. 4g). These results indicate that Wnt signaling promotes active YAP resulting in YAP release from DVL via the alternative Wnt pathway as suggested by Park HW et al.

We agree with the reviewer's point that DVL is critically required for Wnt signaling and is not a tumor suppressor. To avoid reader confusion, we carefully revised our manuscript to note that DVL is a molecular effector of phosphorylated YAP, and added a comment on the essential role of DVL in Wnt signaling in the Discussion section.

It is important to demonstrate that Dvl selectively binds the phosphorylated YAP because phosphorylation is known to play a major role in YAP regulation and the effect of Dvl on YAP appears to be dependent on YAP phosphorylation. A simple in vitro protein-protein interaction will convincingly demonstrate whether Dvl3 binds to phosphorylated or unphosphorylated YAP. The observed increase of Dvl-YAP interaction presented in the manuscript can be easily explained as a consequence of YAP cytoplasmic localization that is caused by YAP phosphorylation, rather than the enhanced affinity of Dvl towards the phosphorylated YAP.

Response) We appreciate the helpful comment regarding a simple in vitro interaction study. Following this suggestion, we have performed an additional in vitro experiment with lambda phosphatase (λ PPase). We harvested cell lysate from YAP transfected cells, and treated the λ PPase to remove phosphate from immunoprecipitated YAP. Then we compared the interaction between YAP and DVL3 in vitro. Consistent with the data in the primary manuscript, λ PPase treatment decreased DVL affinity toward the dephosphorylated YAP (Fig. 1c).

It has been reported by several groups that TEAD functions to retain YAP in the nucleus whereas 14-3-3 functions to retain phosphorylated YAP in the cytoplasm. What is the relative importance of Dvl in YAP subcellular localization when comparing to TEAD and 14-3-3? To really demonstrate a critical role of Dvl in YAP localization, the authors should generate Dvl3 knockout cells, then determine YAP localization under various conditions, such as cell density and energy stress, to unambiguously show that Dvl is indeed important for YAP nuclear export.

Response) In response to the reviewer's concern about the relative importance of 14-3-3 (YWHAZ) and TEAD, we performed further experiments using immunoprecipitation (IP) and immunofluorescence (IF). In accordance with the well-known function of 14-3-3 to retain phosphorylated YAP in cytoplasm, DVL clearly interacted with 14-3-4 (Supple Fig. 2a). However, DVL did not interact with TEAD transcription factor, while the TEAD bound with YAP, which served as a positive control. In our IF study, endogenous TEAD and DVL were mainly localized in nuclear space and cytoplasmic space, respectively, indicating the relative importance of DVL to 14-3-3 (Supple Fig. 2b).

To determine the unambiguous function of DVL3 on YAP localization, we used DVL1/DVL2/DVL3 triple knockout (DVL-TKO) 293 T-REx cells with parental cells (*Paclikova et al, Mol Cell Biol 2017, 37, e00145-17*) and performed further

experiments. Indeed, YAP was predominantly localized in nuclear space in DVL-TKO cells under confluent contact inhibition compared to wt counterpart (Fig. 2f), and re-introduction of DVL3 into DVL-TKO cells successfully decreased nuclear YAP abundance (Supplementary Fig. 4b). We also examined the essential role of DVL using DVL-TKO cells on the YAP localization regulated by metabolic stress and adherens complex (Fig. 5c and Fig. 7b).

Fig.S1b-c. Because different antibodies were used to detect different Dvl proteins, simply based on the Western intensity the authors cannot conclude that Dvl3 is most abundant in these cancer cells.

Response) We agree with the reviewer's concern about the different antibody to determine relative abundance of isoforms. Because there is no commercially available antibody for all DVL isoforms, we have examined relative transcript abundance of DVL isoforms using primers having similar target sizes (80-85 bp). Consistent with TCGA data and western blot analysis, DVL3 transcripts were dominantly abundant in a panel of cancer cell lines (Supplementary Fig. 1b).

Fig.1a. A control, such as DVL3 knockdown or knockout, is needed to show that the Dvl3 antibody is specific in the staining.

Response) We have included the immunofluorescence staining from DVL3 knockdown cells in Supplementary Fig. 1c. See also DVL3 immunofluorescence staining in wt and DVL-TKO 293 T-REx cells (Fig. 2f).

Fig.2c. Cell density has a very important effect on YAP localization. Based on the DNA staining (insert), it appears that the Dox+ cells have lower cell density, which should have more nuclear YAP. It is extremely important to have exactly same cell density in order to look at YAP localization. Does Dvl3 knockdown affect YAP phosphorylation?

Response) We appreciate the reviewer's constructive comments on cell density. Following the reviewer's concern, we have carefully performed further experiments. We have maintained confluent condition and treated the doxycycline to induce two independent shRNA for DVL3 for 48 hr. To validate confluent cell density, we also examined E-cadherin to show tight cell-cell contact and obtained results similar to those in the primary manuscript (Fig. 2c and Supplementary Fig. 4a). We also agree on the issue of YAP phosphorylation status according to DVL abundance that is a concern also raised by Reviewer 2. We further examined phosphorylated YAP to determine whether DVL3 knockdown or knockout affects YAP phosphorylation. Loss-of-DVL3 did not affect YAP phosphorylation in nucleus or in cytoplasm as determined by pSer127-YAP and mobility shift on a phos-tag gel (Fig. 2d, Fig. 2f and Supplementary Fig. 4a). Consistent with previous observations that

Ser127 phosphorylation is critical for cytoplasmic retention of YAP (Zhao et al., *Genes Dev* 2010, 24, 72-85), only active YAP existed in nuclear space regardless of DVL3 abundance. We also added a comment on YAP phosphorylation (such as Ser109 and Ser381) and intracellular dynamics regulated by DVL in the Discussion section. See also our response about YAP phosphorylation to Reviewer 2.

Fig.S4a. The cytoplasmic YAP levels and expression of Dvl3 should be included. It was reported that the PDZ domain in Dvl is required for interaction with TAZ (*Dev. Cell.* 18, 579-591, 2010). Do the authors have explanation why deletion of YAP PDZ domain has no effect on the TEAD reporter?

Response) We agree the reviewer's concern regarding YAP-DVL interaction, an issue also raised by Reviewer 2. To address this issue, we further determined the role of DVL's conserved domains on YAP interaction. In C-terminus of YAP, there is a PDZ binding motif that is important for interaction with YAP-binding partners harboring the PDZ domain, such as ZO (*Biochem J*, 2010, 432, 461-472). When we deleted 5 amino acids (FLTWL) of the PDZ binding motif, the YAP was not able to bind to DVL3 (Supplementary Fig. 5b), indicating that the PDZ domain of DVL3 is required for YAP interaction. Consistent with the provided reference (*Dev. Cell* 1010, 18, 579-591) demonstrating the importance of PDZ and PY (PPxY) motif of DVL to TAZ binding, we found that YAP also binds to PDZ and PY motif on DVL (Supplementary Fig. 5b). The deletion of both PDZ and PY motif on DVL3 abolished YAP interaction as well as YAP cytoplasmic translocation (Supplementary Fig. 5c). The cytoplasmic YAP and DVL3 levels (in Supplementary Fig. 4a in primary manuscript) were included following the reviewer's suggestion (Supplementary Fig. 5a in revised manuscript). See also our response to Reviewer 2.

Fig.3c. The experimental conditions could be problematic. For the un-transfected cells (no red color), YAP was cytoplasmic in the Dvl3-wt panel but more nuclear YAP in the Dvl3-ASA-mut panel. Therefore, the difference could largely be due to the difference in culture conditions, which would affect YAP localization. Can Dvl3 expression induce YAP cytoplasmic localization in LATS1 and LATS2 knockout cells?

Response) Following the reviewer's concern, we have generated stable cell lines to avoid transfection and culture condition issues. Now we have clearer results on ASA nuclear export mutant compared to wild type of DVL3 in MCF-7 and MCF-10A cells (Fig. 2c and Supplementary Fig. 5e).

To address the DVL's role on YAP localization in LATS knockout cells, we have performed the suggested experiments in LATS1/2-double knockout (*Lats1/2^{-/-}*) 293A and mouse embryonic fibroblast (MEF) cells (kindly provided by Park HW at Yonsei University). As shown in Fig. 4d and Supplementary Fig. 6 in the revised manuscript, DVL3's role was defective in terms of YAP cytoplasmic

localization in *Lats1/2*^{-/-} 293A and MEF cells although the DVL3 remained in cytoplasmic space, supporting the importance of LATS kinases on YAP nuclear export by DVL.

Fig.5a. Key controls of YAP localization without transfection of E-cadherin or α -catenin need to be included. 5b. YAP levels in the cytoplasmic fraction should be included. 5c. Is the color for HECD antibody (red, indicated by arrows) same as YAP (red)?

Response) Following the reviewer's concern, we have added control of YAP localization without transfection of E-cadherin or α -catenin in Fig. 5a and Supplementary Fig. 7a. We re-examined the YAP levels with E-cadherin and cadherin or α -catenin, and added YAP levels in whole cell lysates as also requested by Reviewer 2 (Fig. 5b and Supplementary Fig. 7b). We also performed an immunofluorescence study again with a different color for HECD antibody (Fig. 5c).

Fig.7a-c. Both AMPK and LKB1 knockout cells are widely available. Why the authors used overexpression of dominant negative mutants to do these experiments instead of the much cleaner experiments with knockout cells? The data in 7a is strange. Overexpression of AMPK-KD or LKB1-KD caused a dramatic dephosphorylation and increased nuclear localization, why the TEAD reporter activity was not increased by the dominant negative AMPK or LKB1?

Response) Following the reviewer's suggestion, we have obtained wt and AMPK α 1/ α 2 double-knockout (DKO) MEF (mouse embryonic fibroblast) cells (kindly provided by Park HW at Yonsei University), and performed western blot and immunofluorescence study. Consistently, we found that DVL did not translocate YAP into cytoplasm in AMPK DKO MEF cells compared to wild type cells (Fig. 7d). To determine the role of endogenous LKB1 on DVL's intracellular YAP localization, we chose A549 non-small cell lung cancer (NSCLC) cells, a well-characterized cell line of LKB1-deficient having wild type p53 status. Consistent with the well-described role of the LKB1-AMPK axis in human cancer and AMPK-mediated YAP phosphorylation (*Shackelford DB & Shaw RJ, Nat Rev Cancer 2009, 9, 563-575; Mo J-S et al, Nat Cell Biol 2015, 17, 500-510*), endogenous YAP was not translocated into cytoplasm under FBS-deprived confluent culture condition (Supplementary Fig. 9c). Further, overexpression of DVL3 was not able to translocate YAP into cytoplasmic space regardless of sparse or confluent culture condition (Fig. 7d), supporting the importance of LKB1 tumour suppressor on DVL's YAP localization. We appreciate the reviewer's pointing out the normalization error in TEAD reporter assay. We have re-examined the role of AMPK-KD or LKB-KD on TEAD reporter activity regulated by DVL3 and added new data of relative fold repression on Fig. 7c in the revised manuscript. To avoid confusion, we also added data on

relative TEAD reporter activity by AMPK-KD and LKB-KD in Supplementary Fig. 9a.

7d. An essential control of LKB1-KD is missing. Moreover, this data is not sufficient to conclude that LKB1-KD function through YAP to promote tumor growth.

Response) Following the reviewer's suggestion, we have repeated in vivo experiments with vector or LKB-KD control. We have similar results to the primary manuscript indicating that co-activation of canonical Wnt by DVL and YAP activity under LKB-KD context significantly promote tumourigenic potential (Fig. 7f).

7e. This manuscript provides insufficient data for the model, which is rather speculative.

Response) Following the reviewer's concern, we have moved our proposed model to Supplementary Fig. 8d.

Page 11, line 236, typo "hemophilic"

Page 15, line 326, should be Fig.7d. Fig 7c is not cited in the text.

Response) We have corrected the mistakes in the revised manuscript.

Reviewer #2:

Although the phosphorylation dependent cytoplasmic localization of YAP/TAZ has been widely studied, the key effector molecule(s) that determine the nucleocytoplasmic shuttling of YAP/TAZ have not been well investigated. This manuscript describes a regulation of YAP activity by DVL3, a known component of the Wnt signaling pathway. Cytoplasmic YAP/TAZ has been shown to restrict Wnt signaling by limiting the activity (Varelas, 2010) and nuclear localization of DVL (Barry, 2013). Reciprocal to these findings, in the current manuscript Lee et al studied the regulation of YAP activity by DVL. The authors show that DVL3 is the nucleocytoplasmic shuttling effector of phosphorylated YAP which is induced by upstream signaling such as E-cad, α -cat and AMPK via Lats. Depletion of DVL3 or mutation of the NES of DVL3 (DVL3-ASA) increases the nuclear level of YAP. This is an important new contribution to the field.

Authors also claim that the dysregulation of p53 or LKB1 signaling pathway in breast cancer relieves the restriction of YAP activity by DVL3, and leads to the co-activation of TEAD and TCF/LEF transcriptional activities. This suggests the tumor suppressor role of p53 and LKB1 signaling through the regulation of YAP activity by DVL3.

Overall, the manuscript is well written and the conclusion of this manuscript is supported by experimental results. However, the authors should provide more experimental results and/or discussion to support their proposed model.

1. In many places the authors suggest that YAP is regulated by the Wnt pathway, but this has not yet been shown. DVL is a component of the Wnt pathway, but it is possible that it has a distinct function in the Hippo pathway. This is an important question and the authors should test whether stimulation with a soluble Wnt affects YAP through this mechanism. It doesn't matter which result they get, but they should perform the experiments and discuss this issue.

Response) We appreciate the critical comments on our manuscript. Previously, Park HW et al reported that soluble Wnt activates YAP through alternative Wnt signaling (Park HW et al, Cell 2015, 162, 780-794). To address this issue, we have treated soluble Wnt1 or Wnt3a and examined the YAP phosphorylation status. Consistently, soluble Wnt1 and Wnt3a increased unphosphorylated active YAP while total YAP level was unaffected (Fig. 4f). In this setting, molecular interaction between YAP and DVL3 was decreased by the Wnt ligands. In immunofluorescence study, soluble Wnt1 and Wnt3a induced YAP nuclear localization while the DVL3 was largely retained in cytoplasmic space (Fig. 4g). These results indicate that Wnt ligands activate YAP (decreased YAP phosphorylation) and YAP is subsequently released from DVL. Regarding the complexity and importance of DVL's dynamics within the Wnt pathway, we also added comments in the Discussion section. We thank the reviewer again for the suggested critical experiments with soluble Wnt.

2. Varelas et al (Dev cell 2010; 18(4) 579–591) identified the TAZ binding domain in DVL2. Rescue of DVL3 depleted cells with a mutant defective for YAP binding followed by immunofluorescence staining of YAP would add further support of DVL3 as a shuttling effector of YAP.

Response) We agree with the reviewer's point regarding YAP binding domain in DVL, also raised by Reviewer 1. To address this issue, we further examined the role of conserved domains of DVL and YAP in molecular interaction. We first examined YAP interaction with deletion mutants of conserved domains (DIX or PDZ or DEP) in DVL. Interestingly the deletion of conserved domain in DVL decreased the YAP interaction, but those mutants still harbored YAP binding activity and suppressor function of nuclear YAP (Supplementary Fig. 5a), indicating that YAP interacts with DVL via multiple domains similarly to TAZ (Varelas et al, Dev Cell 2010, 18, 579). Second, the PDZ-binding motif located in C-terminus of YAP is indispensable for the binding to other partner proteins, such as ZO-2 (Biochem J, 2010, 432, 461-472). Therefore, we made a C-terminal deletion mutant of YAP to examine whether the PDZ-binding motif of YAP is required for interaction with DVL. Indeed, deletion of the PDZ-binding

motif consisting of 5 amino acids (PFTWL) of YAP is sufficient to ablate DVL interaction (Supplementary Fig. 5b), indicating that the PDZ domain in DVL is important for YAP interaction. The PY motif (PPxY) in YAP binding partners, such as LATS and AMOT, is critically important to bind YAP (Zhao B. et al, *Genes Dev* 2011, 25, 51-63; Meng Z. et al, *Genes Dev* 2016, 30, 1-17). Based on these observations, we next made mutant expression vector deleting both the PDZ domain and PY (PPxY) motif of DVL and examined the binding affinity to YAP. Indeed, the PDZ and PY motifs of DVL are required for YAP binding (Supplementary Fig. 5b). Further, the mutant defective DVL for YAP binding was unable to affect YAP cytoplasmic translocation in DVL3 depleted cells (Supplementary Fig. 5c). See also our response to Reviewer 1.

3. The reports by Mana-Capelli et al (*Mol Biol Cell*. 2014;25(10):1676-85) and Moleirinho et al (*Mol Biol Cell*. 2014;25(10):1676-85) should be mentioned and discussed as they showed the cytoplasmic retention of YAP by angiotensins.

Response) We thank the reviewer for interesting and valuable references relevant to our study. We have mentioned and discussed suggested reports about angiotensin (AMOT) in detail.

4. The authors should show the phosphorylation status of nuclear YAP in DVL3 depleted cells (Fig 2b, 2c, Sup Fig 3) and DVL3-ASA mutant expressing cells (Fig 3d). The decrease in YAP phosphorylation by the inactivation of upstream kinases enriches the level of nuclear YAP protein. If DVL3 plays a role as a nucleocytoplasmic shuttling effector of YAP without regulating the upstream kinases of YAP, depletion of DVL3 or overexpression of DVL3-ASA should increase the level of phosphorylated YAP in the nucleus.

Response) We agree regarding critical issue of nuclear YAP phosphorylation status according to the DVL abundance. To address this issue, we have performed further experiments with pSer-127-YAP antibody and phos-tag gel. Interestingly, DVL abundance and DVL3-ASA mutant did not affect the YAP phosphorylation status in nuclear and cytoplasmic space (Fig. 2b, Fig. 2d, Fig. 3d and Supplementary Fig. 4a), indicating that DVL only exports phosphorylated YAP from nucleus. Interestingly, increased nuclear YAP abundance by depletion of DVL3 remained unphosphorylated (Fig. 2d, Fig. 2f and Supplementary Fig. 4a), suggesting the potential role of DVL in YAP phosphorylation in nuclear space. While we mainly focused on DVL's YAP nuclear export in this study, dynamic shuttling of DVL coupled with multiple YAP phosphorylation cascade in nuclear space needs further study. We also have added comments on DVL's role in YAP phosphorylation on other sites (such as Ser109 and/or Ser381), probably linked to the AMOT-Lats-YAP complex, in the Discussion section of the revised manuscript.

5. Fig 4b and 5b needs Western blot results of YAP in the whole cell lysates for the

proper quantification.

Response) We have added western blot of YAP in the whole cell lysates. We quantified the relative nuclear YAP level comparing nuclear loading control (HDAC1) of same loading samples rather than whole cell lysates.

6. Authors should include the immunofluorescence staining results of control 293 cells in Fig 5a.

Response) We have added the immunofluorescence results to address the reviewer's concern (Fig. 5a and Supplementary Fig. 9b).

There are numerous typos and errors throughout the manuscript:

1. DLV should be corrected to DVL (line #137, #313)
2. Misspelling of suppresses (line #122)
3. Fig. 7c at line 326 should refer to Fig. 7d.
4. Hemophilic should be corrected to homophilic (line #236, #775)
5. Typo of E-cadherin (line #238)
6. 293T should be changed to 293 (line #423, #532), as the entire manuscript described that authors used 293 cells.
7. There is a labeling error in Fig 2c.

Response) Thank you for careful reading of our manuscript. We have corrected the mistakes in the text.

- END -

Reviewers' comments:

Reviewer #2 (Remarks to the Author):

The authors responded to reviewers' comments satisfactorily and changed the manuscript accordingly by the addition of many new experiments and controls. The new data support their previous conclusions.

A minor correction is needed for the sentence on page 13, line 273 "Consistent with YAP phosphorylation-dependent binding to DVL, soluble Wnt ligands treatment decreased DVL-YAP binding affinity". The authors concluded this because DVL doesn't bind to YAP-5SA, and soluble Wnt decrease YAP phosphorylation and induces nuclear YAP localization. However, they didn't provide direct evidence that Wnt ligands change the binding "affinity" of DVL-YAP. They should remove "affinity" from the sentence to clarify.

Reviewer #3 (Remarks to the Author):

The authors have failed to adequately address the major concerns of reviewer 1. The problem with YAP translocation in cell culture is that it is highly sensitive to cell density, which is hard to perfectly control for. Thus, all of the effects they see on YAP localisation could be indirectly caused by changes in density.

If the authors are confident of their model, why not look at DVL knockout mice, zebrafish or Drosophila to demonstrate nuclear trapping of YAP/Yki in vivo? Without in vivo data, the entire manuscript remains questionable.

Reviewer #4 (Remarks to the Author):

The Hippo signaling is an evolutionary conserved pathway that inhibits cell proliferation by contact inhibition, and whose loss leads to organ growth and cancer development. The Yes-associated protein (YAP) transcription co-activator is a key regulator of the Hippo pathway. Inhibition of the Hippo pathway leads to increased nuclear YAP abundance and TEAD transcriptional activity resulting in increased organ size as well as overgrowth of cancer. Therefore it is important to understand the mechanisms of YAP translocation between cytoplasm and nuclear. Authors showed Wnt scaffolding protein Dishevelled (DVL) is responsible for cytosolic translocation of phosphorylated YAP. In addition, authors showed DVL is also required for YAP subcellular localization induced by E-cadherin, α -catenin, or AMPK activation. Furthermore, authors showed the nuclear-cytoplasmic trafficking is largely dependent on the p53-Lats2 or LKB1-AMPK tumor suppressor axes.

Overall, the manuscript is well written and the conclusion of this manuscript is supported by experimental results. Authors have addressed most previous reviewers' comments. However, there were two major concerns about the author proposed model:

1) Authors demonstrated the interaction between DVL and YAP is through PDZ-/PPxY- domains of DVL and PDZ-binding domain of YAP. On the other hand, authors also showed DVL and YAP interaction depends on YAP phosphorylation. There is no YAP phosphorylation site located in YAP PDZ-binding domain. Authors did not provide any insight about this point.

2) Authors showed NES mutant of DVL retained YAP in the nuclear by immunostaining and YAP targets expression, however, authors did not show whether DVL and YAP interacted in the nuclear under this condition, again there is no evidence that YAP can be phosphorylated in the nuclear, how authors explain the contradiction?

Some minor points are as below:

1)Supplemental Figure 5a data is confusing, the authors claimed (line 191-193): "deletion mutants of those conserved domains suppressed nuclear YAP abundance and TEAD transcriptional activity, and the deletion mutants of DVL3 retained binding ability to YAP". However, the data showed that deletion mutants of DVL3 dramatically disrupted the interaction between YAP and DVL3 (supplemental Fig 5a. right panel).

2)There is contradiction in supplemental Figure 5b data, authors showed deletion of PDZ binding domain of YAP completely disrupted YAP and DVL3 interaction, indicating WW domain of YAP does not involve in YAP and DVL3 interaction (supplemental 5b middle panel). On the other hand, authors showed YAP and DVL3 interaction can be disrupted only under the condition of both DVL3 PDZ and PPxY domains (WW domain binding ligand) deletion (supplemental Fig 5b right panel).

3)It has been extensively reported that YAP-S127A predominantly localized in nuclear, however, a substantial amount of YAP-S127A localized in cytoplasm (Figure 4b).

Response to Reviewers (NCOMMS-17-16923-T)

We appreciate the reviewers' helpful comments and constructive criticisms on our manuscript. We have performed experiments to address concerns raised by reviewers. Our responses to the reviewers are provided in bold below.

Reviewers' comments:

Reviewer #2 (Remarks to the Author):

The authors responded to reviewers' comments satisfactorily and changed the manuscript accordingly by the addition of many new experiments and controls. The new data support their previous conclusions.

A minor correction is needed for the sentence on page 13, line 273 "Consistent with YAP phosphorylation-dependent binding to DVL, soluble Wnt ligands treatment decreased DVL-YAP binding affinity". The authors concluded this because DVL doesn't bind to YAP-5SA, and soluble Wnt decrease YAP phosphorylation and induces nuclear YAP localization. However, they didn't provide direct evidence that Wnt ligands change the binding "affinity" of DVL-YAP. They should remove "affinity" from the sentence to clarify.

Authors' Response) We appreciate the helpful comment and have removed the word "affinity" in the revised manuscript following the suggestion.

Reviewer #3 (Remarks to the Author):

The authors have failed to adequately address the major concerns of reviewer 1. The problem with YAP translocation in cell culture is that it is highly sensitive to cell density, which is hard to perfectly control for. Thus, all of the effects they see on YAP localisation could be indirectly caused by changes in density.

If the authors are confident of their model, why not look at DVL knockout mice, zebrafish or Drosophila to demonstrate nuclear trapping of YAP/Yki in vivo? Without in vivo data, the entire manuscript remains questionable.

Authors' Response) The experimental cell culture procedures for YAP cytoplasmic translocation induced by confluent contact inhibition and serum starvation (as described in the Figure Legends in our manuscript) have been well-established by other scientists (Zhao B, et al, 2007, 21, 2747-2761; Kim N-G, et al, PNAS 2011, 108, 11930; Fan R, et al, PNAS 2013, 110, 2569-2574). While the DVL knockout mice model may strongly support our claims, it should be noted that three isoforms of murine Dvl gene are broadly expressed during

embryonic development and in adult tissue and that double knockout (such as Dvl1/Dvl2 or Dvl2/Dvl3) mice are embryonic lethal (Gao C. & Chen Y-G. Cell Sig 2010, 22, 717-727). To address the reviewer's concern about tight contact inhibition and in vivo evidence, we have added in vivo data showing that induction of wt or ASA mutant DVL differentially localized YAP in cells. We examined YAP localization from xenografted 293 cells having Tet-inducible wt or ASA mutant DVL. As shown Fig. 3h in the revised manuscript, endogenous YAP was differentially localized in vivo tumour tissue where the cells have tight 3-dimensional contact.

Reviewer #4 (Remarks to the Author):

The Hippo signaling is an evolutionary conserved pathway that inhibits cell proliferation by contact inhibition, and whose loss leads to organ growth and cancer development. The Yes-associated protein (YAP) transcription co-activator is a key regulator of the Hippo pathway. Inhibition of the Hippo pathway leads to increased nuclear YAP abundance and TEAD transcriptional activity resulting in increased organ size as well as overgrowth of cancer. Therefore it is important to understand the mechanisms of YAP translocation between cytoplasm and nuclear. Authors showed Wnt scaffolding protein Dishevelled (DVL) is responsible for cytosolic translocation of phosphorylated YAP. In addition, authors showed DVL is also required for YAP subcellular localization induced by E-cadherin, α -catenin, or AMPK activation. Furthermore, authors showed the nuclear-cytoplasmic trafficking is largely dependent on the p53-Lats2 or LKB1-AMPK tumor suppressor axes.

Overall, the manuscript is well written and the conclusion of this manuscript is supported by experimental results. Authors have addressed most previous reviewers' comments. However, there were two major concerns about the author proposed model:

1) Authors demonstrated the interaction between DVL and YAP is through PDZ-/PPxY- domains of DVL and PDZ-binding domain of YAP. On the other hand, authors also showed DVL and YAP interaction depends on YAP phosphorylation. There is no YAP phosphorylation site located in YAP PDZ-binding domain. Authors did not provide any insight about this point.

Authors' Response) We appreciate the critical comment regarding YAP and DVL interaction although our primary manuscript mainly focused on the YAP nuclear export function of DVL and its functional relevance in tumour suppressor context. To address the reviewer's concern about YAP, we further characterized the PDZ-binding domain located in c-terminus of YAP. Previously, SET7 (SEDT7)-mediated monomethylation of lysine 494 of YAP, where it closely locates in the PDZ-binding domain, had been identified as critical for

cytoplasmic retention despite normal S127 phosphorylation (Oudohoff MJ et al, Dev Cell 2013, 26, 188-194). To examine the role of the monomethylation modification on PDZ-binding domain, we made a point mutant in which lysine (K) was substituted with arginine (R) on YAP, and subjected the lysate to immunoprecipitation. Interestingly, K494R point mutation of YAP was sufficient to abolish DVL3 binding similar to deletion of the PDZ-binding domain (Supplementary Fig. 5b). These results suggest that (1) monomethylation of K494 may cooperate with PDZ-binding domain to interact with DVL, and (2) SET7-mediated methylation of nonhistone substrates may be closely related to YAP phosphorylation dynamics and DVL-mediated YAP nuclear export. Further study is required for structural and biological characterization of monomethylation and PDZ-binding domain in YAP and other proteins. We added the relevant comments in the revised manuscript. See also our response to WW domain of YAP in below.

2) *Authors showed NES mutant of DVL retained YAP in the nuclear by immunostaining and YAP targets expression, however, authors did not show whether DVL and YAP interacted in the nuclear under this condition, again there is no evidence that YAP can be phosphorylated in the nuclear, how authors explain the contradiction?*

Authors' Response) We agree with the reviewer's point, which is critically important for our claims. To address the reviewer's concern, we have performed immunoprecipitation assay with nuclear fraction of NES mutant expressing samples. We have adjusted input YAP abundance from nuclear protein of NES mutant and from whole cell lysate of wt DVL3 as a positive control for YAP binding. Interestingly, NES mutant of DVL did not interact with nuclear YAP that is unphosphorylated (Fig. 3e), indicating that YAP and NES mutants do not interact in nuclear space.

Some minor points are as below:

1) *Supplemental Figure 5a data is confusing, the authors claimed (line 191-193): "deletion mutants of those conserved domains suppressed nuclear YAP abundance and TEAD transcriptional activity, and the deletion mutants of DVL3 retained binding ability to YAP". However, the data showed that deletion mutants of DVL3 dramatically disrupted the interaction between YAP and DVL3 (supplemental Fig 5a. right panel).*

Authors' Response) We have performed repeated experiments to determine interaction between YAP and deletion mutants of DVL. We have revised data that is consistent with nuclear YAP abundance and TEAD reporter activity (Supplemental Fig. 5a).

2) *There is contradiction in supplemental Figure 5b data, authors showed deletion of*

PDZ binding domain of YAP completely disrupted YAP and DVL3 interaction, indicating WW domain of YAP does not involve in YAP and DVL3 interaction (supplemental 5b middle panel). On the other hand, authors showed YAP and DVL3 interaction can be disrupted only under the condition of both DVL3 PDZ and PPxY domains (WW domain binding ligand) deletion (supplemental Fig 5b right panel).

Authors' Response) While we did not examine the role of WW domain of YAP in primary manuscript, we have now further characterized the WW domain of YAP in DVL interaction. We made deletion mutant of the WW domain of YAP and examined interaction with DVL. Deletion of the WW domain of YAP largely abolished DVL interaction (Supplemental Fig. 5b), supporting that WW domain of YAP interacts with PY motif in DVL (Chen HI & Sudol M, PNAS 1995, 92, 7819-7823; Macias, MJ et al, Nature 1996, 382, 646-649). We also added a description of the importance of the WW domain in the revised manuscript.

3) It has been extensively reported that YAP-S127A predominantly localized in nuclear, however, a substantial amount of YAP-S127A localized in cytoplasm (Figure 4b).

Authors' Response) We agree with the reviewer's concern regarding phosphorylation-resistant Yap mutants' YAP localization. We interpreted the cytoplasmic localization of the mutant as an issue of overexpression experiment because over-expression of phosphorylation-resistant YAP in culture cells and tissue of transgenic mice are not restricted to the nucleus (Zhao B. et al, Genes Dev 2007, 21, 2747-2761, Fig. 4; Barry et al, Nature 2013, 493, 106-110, Supplementary Fig. 1a). To avoid readers' confusion, we transfected a minimum amount (50 ng) of YAP expression vectors and examined YAP localization with confocal microscopy. We now show YAP phosphorylation mutants are mainly localized in nucleus independent of DVL (Fig. 4b).

END

REVIEWERS' COMMENTS:

Reviewer #4 (Remarks to the Author):

The authors have made a great effort to address the points raised in the review, and I think the paper is now acceptable for publication.